# Dysbiosis of human gut microbiome in young-onset colorectal cancer

Yongzhi Yang[1,2,7], Lutao Du[3,7], Debing Shi[1,2,7], Cheng Kong[4,7], Jianqiang Liu[5,7], Guang Liu[6,7], Xinxiang Li[1,2] & Yanlei Ma [1,2✉]

The incidence of sporadic young-onset colorectal cancer (yCRC) is increasing. A significant knowledge gap exists in the gut microbiota and its diagnostic value for yCRC patients. Through 16S rRNA gene sequencing, 728 samples are collected to identify microbial markers, and an independent cohort of 310 samples is used to validate the results. Furthermore, species-level and functional analysis are performed by metagenome sequencing using 200 samples. Gut microbial diversity is increased in yCRC. *Flavonifractor plautii* is an important bacterial species in yCRC, while genus *Streptococcus* contains the key phylotype in the old-onset colorectal cancer. Functional analysis reveals that yCRC has unique characteristics of bacterial metabolism characterized by the dominance of DNA binding and RNA-dependent DNA biosynthetic process. The random forest classifier model achieves a powerful classification potential. This study highlights the potential of the gut microbiota biomarkers as a promising non-invasive tool for the accurate detection and distinction of individuals with yCRC.

[1] Department of Colorectal Surgery, Fudan University Shanghai Cancer Center, Shanghai, China. [2] Department of Oncology, Shanghai Medical College, Fudan University, Shanghai, China. [3] Department of Clinical Laboratory, The Second Hospital of Shandong University, 250033 Jinan, Shandong province, China. [4] Department of GI Surgery, Shanghai Tenth People's Hospital Affiliated to Tongji University, Shanghai, China. [5] Department of Endoscopy, Fudan University Shanghai Cancer Center, Shanghai, China. [6] Quantum Hi-Tech Microecological Medical (Guangdong) Co.,Ltd., Guangzhou, China. [7]These authors contributed equally: Yongzhi Yang, Lutao Du, Debing Shi, Cheng Kong, Jianqiang Liu, Guang Liu. ✉email: yanleima@fudan.edu.cn

Colorectal cancer (CRC) is the third most common cancer worldwide, affecting ~10% of patients under 50 years of age[1]. As a traditional elderly disease, CRC incidence in the young is steadily rising across the globe[2–4]. In contrast, the incidence of CRC in older patients is seeing a progressive decrease in the developed world, which is likely to be attributed to population-based CRC screening[5,6]. Young-onset CRC (yCRC) patients often present with more advanced disease and adverse pathological features compared to their older counterparts[7]. This may have a negative impact on their survival outcome[8,9]. Diagnostic and therapeutic protocols dedicated to sporadic CRC in young individuals are currently an unmet clinical need. Also, there is no consensus on whether yCRC patients are indistinguishable or different omics entities compared to old-onset CRC (oCRC) patients.

Evidence is accumulating that the intestinal microbiota, harboring far more genes than our human genome, has emerged as a key environmental factor implicated in the development of CRC, especially within the different ages demographic[10,11]. The potential of the gut microbiota to affect health is particularly relevant for older or younger individuals because the microbiota may modulate aging-related changes in innate immunity, inflammation, and cognitive function[12,13]. Both cell culture-dependent and -independent studies have shown that the gut microbiota of the elderly is different from that of the young[14,15]. Zhang et al. tested the fecal samples of 314 young people from multi-ethnic and multi-regional areas of China and found a functional core group of gut microbiota in a healthy young Chinese population, including a variety of bacteria that participate in the production of short-chain fatty acids, maintaining intestinal mucosal barrier and anti-inflammatory function in healthy individuals[16]. However, major risk factors (unhealthy diets, obesity, and sedentary lifestyles) are becoming more prevalent in successive generations, raising the question of whether altered gut microbiota—especially in the early years of life—could interact with an underlying genetic backdrop to trigger the early onset of the disease. Studies have reported that under long-term physiological stress, the composition and metabolic changes of young people's gut microbiota are consistent with the increase in intestinal permeability and inflammation[17,18]. Specific gut bacteria can invade at least half of the colonic mucus in patients with CRC but without a predisposition for hereditary disease[19]. Therefore, there may exist a characteristic pathogenic bacteria spectrum with diagnostic value in yCRC.

In the present study, a total of 1038 samples are submitted to 16S rRNA gene sequencing, and 200 samples are analyzed by metagenome sequencing. The fecal microbial composition, the functional changes of the microbial communities and the microbial markers of yCRC are explored. We hypothesize that the yCRC and oCRC may have different gut microbial bases, which may shed light on the pathogenesis of CRC at different ages, and may serve as a promising non-invasive biomarker for sporadic yCRC.

## Results

**Participate information and study design.** In total, 1038 eligible cases including 185 yCRC, 379 oCRC, 217 age-matched healthy controls for the yCRC (yControl) and 257 age-matched healthy controls for the oCRC (oControl) were included in this study according to the strict recruitment process (Fig. 1). The fecal microbiota was assessed using 16S rRNA gene sequencing ($n = 728$ from the Fudan cohort and $n = 310$ from the Huadong cohort) and metagenomic sequencing ($n = 200$). Subsequently, the Fudan cohort was randomly assigned to the training phase (Accounted for 70%) and the testing phase (Accounted for 30%).

In the training phase, the microbial markers and classifier were identified by random forest model between CRC and age-matched control (163 oCRC vs. 142 oControl; 100 yCRC vs. 104 yControl). In the testing phase, 70 oCRC, 44 yCRC, 61 oControl, and 44 yControl were recruited to validate the strength of microbial classifier for distinguishing oCRC or yCRC from health. Moreover, to further measure the strength of observed associations, 41 yCRC, 146 oCRC, 54 oControl, and 69 yControl from Huadong cohort were served as independent external validation phase. Non-quantitative fecal occult blood test (FOBT), serum carcinoembryonic antigen (CEA), and serum carbohydrate antigen 19-9 (CA19-9) were used to compare with the CRC classifier.

The clinical characteristics including age, gender, tumor location, tumor size, tumor differentiation, AJCC disease stage, KRAS/NRAS/BRAF mutation status, non-quantitative FOBT, serum CEA and CA19-9 were matched between oCRC group and yCRC group. The CRC group and its corresponding control group was age-matched, respectively, and there was no statistical difference. Specifically, the Fudan cohort included 728 patients, with mean age of $63.23 \pm 8.56$ (25–75% percentile, 55–69) years in oControl, $64.26 \pm 8.68$ (25–75% percentile, 57–70) years in oCRC, $39.76 \pm 6.11$ (25–75% percentile, 35–45) years in yControl, and $40.45 \pm 7.02$ (25–75% percentile, 36–46) years in yCRC, respectively. The Huadong cohort included 310 patients, with mean age of $60.46 \pm 6.94$ (25–75% percentile, 54–65) years in oControl, $62.42 \pm 7.67$ (25–75% percentile, 55–68) years in oCRC, $37.74 \pm 6.19$ (25–75% percentile, 33–42) years in yControl, and $39.68 \pm 7.11$ (25–75% percentile, 33–45) years in yCRC, respectively. The positive rate of non-quantitative FOBT and serum tumor markers in the CRC group were significantly higher than those in the healthy control group. Detailed clinical data for studied individuals was shown in Supplementary Tables 1, 2 and 3.

**Gut microflora dysbiosis in yCRC.** After denoising using DADA2, an average of 28,455 16S rRNA gene sequences per sample was obtained (min: 4794; max: 143,517; median: 23,660), and 40,032 amplicon sequence variants (ASVs) were obtained from Fudan cohort (Supplementary data 1). To assess the differences of bacterial diversity among groups, sequences were aligned for alpha-diversity. The results showed that fecal microbial alpha-diversity was significantly decreased in the oCRC group compared to oControl group, and the similar decline was found in yCRC compared to yControl. In contrast, microbial diversity was markedly increased in yCRC versus oCRC (Fig. 2a, b). Moreover, a Venn diagram of bacteria showed that 326 of the total 822 genera were shared among the four groups, while 361 of 582 genera were shared between the yCRC group and oCRC group. Notably, 12 of 397 ASVs were unique for yCRC group (Fig. 2c). To display microbiome space between samples, beta diversity was calculated using weighted UniFrac method, and principal coordinates analysis (PCoA) was performed. The results presented a significantly different distribution among groups using permutational multivariate analysis of variance (PERMANOVA) analysis (Fig. 2d). These results suggest that the yCRC group has unique diversity and microbial distance metric from the oCRC group.

**Phylogenetic profiles of fecal microbial communities in yCRC.** To identify differentially abundant taxa in oCRC and yCRC, we performed linear discriminant analysis coupled with effect size analysis (LEfSe) algorithms on fecal microbiota composition between CRC (oCRC or yCRC) and age-matched healthy control (oControl or yControl) based on the 16S rRNA gene sequencing. There were 38 bacterial taxa showing distinct relative abundances between oCRC and oControl, and 24 bacterial taxa were differentially abundant between yCRC and yControl

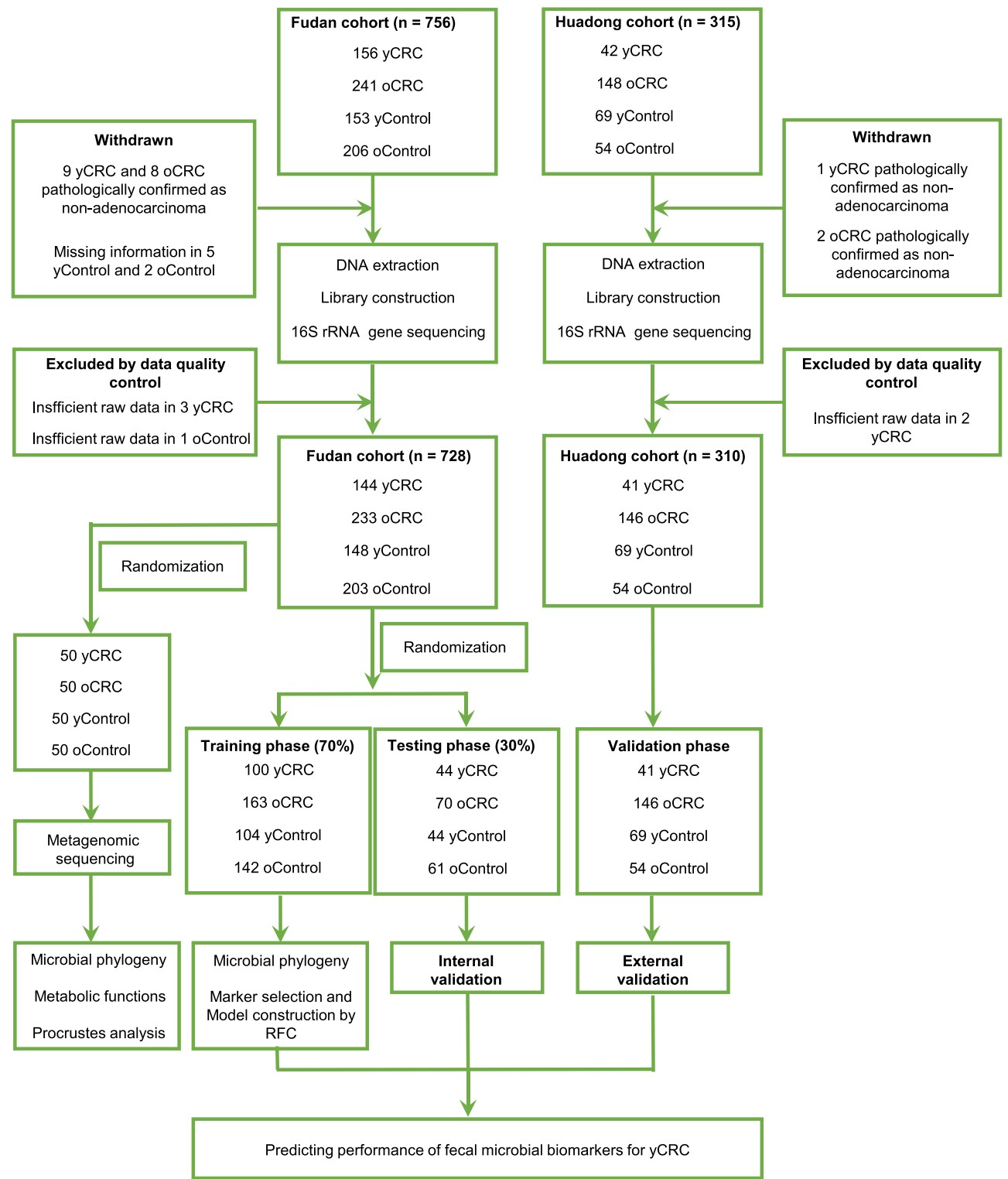

**Fig. 1 Study design and flow diagram.** A total of 1038 eligible cases were included in this study. The fecal microbiota was assessed using 16S rRNA gene sequencing or metagenomic sequencing. 144 yCRC patients, 233 oCRC, and 351 age-matched healthy controls from the Fudan cohort were randomly divided into the training phase (accounted for 70%) and the testing phase (accounted for 30%) to identify the gut microbiome community and microbial markers. The strength of observed associations of microbial markers was further independently verified in 41 yCRC, 146 oCRC, and 123 age-matched healthy controls from the Huadong cohort. CRC colorectal cancer, yCRC young-onset CRC, oCRC old-onset CRC, yControl age-matched healthy controls for the yCRC, oControl age-matched healthy controls for the oCRC, RFC random forest classifier.

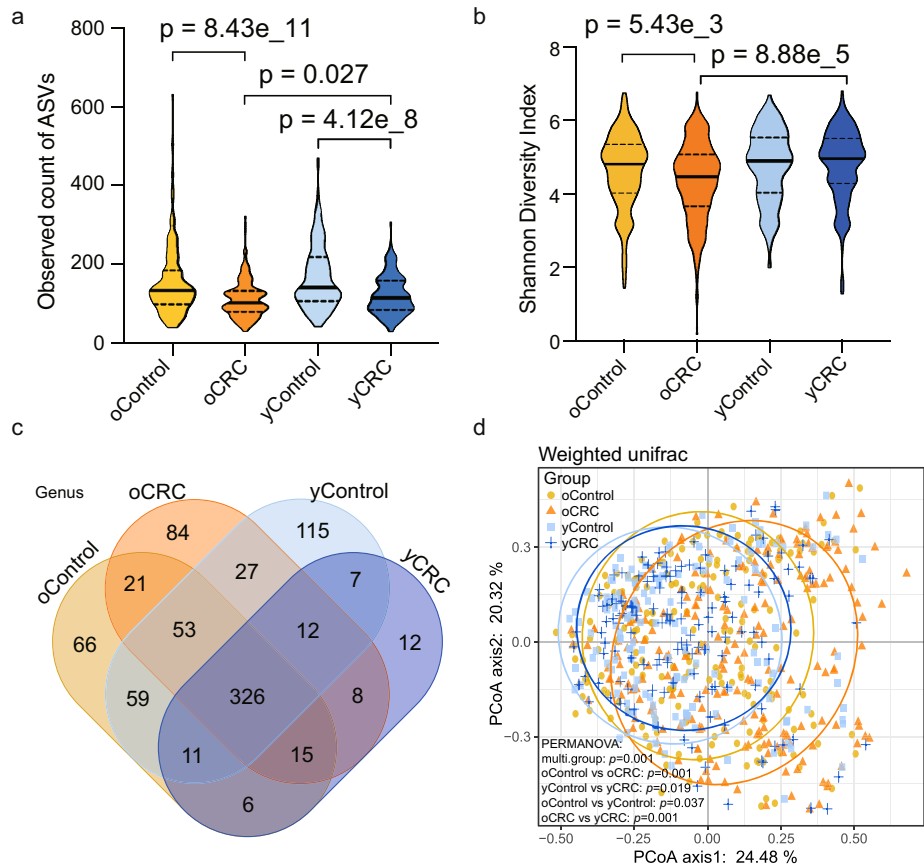

**Fig. 2 Bacterial diversity of the fecal microbiota associated with yCRC and oCRC. a** Fecal microbial diversity estimated by observed count of ASVs. $n = 728$ from Fudan cohort ($n = 233$ for oCRC and $n = 203$ for oControl; $n = 144$ for yCRC and $n = 148$ for yControl). $p$ values are calculated by two-sided unpaired Student's $t$ test. Bars represent standard deviation. **b** Fecal microbial diversity estimated by Shannon index. $n = 728$ from Fudan cohort ($n = 233$ for oCRC and $n = 203$ for oControl; $n = 144$ for yCRC and $n = 148$ for yControl). $p$ values are calculated by two-sided unpaired Student's $t$ test. Bars represent standard deviation. **c** A Venn diagram displayed the overlaps among groups. $n = 728$ from Fudan cohort ($n = 233$ for oCRC and $n = 203$ for oControl; $n = 144$ for yCRC; and $n = 148$ for yControl). **d** Beta diversity calculated by PCoA of weighted UniFrac distances and PERMANOVA. $n = 728$ from Fudan cohort. 203 oControl vs. 233 oCRC, two-sided, $p = 0.001$; 148 yControl vs. 144 yCRC, $p = 0.019$; 203 oControl vs. 148 yControl, $p = 0.037$; 233 oCRC vs. 144 yCRC, $p = 0.001$. CRC colorectal cancer, yCRC young-onset CRC, oCRC old-onset CRC, yControl age-matched healthy controls for the yCRC, oControl age-matched healthy controls for the oCRC, PCoA principal coordinate analysis, PERMANOVA permutational multivariate analysis of variance analysis. Source data are provided as a Source Data file.

(linear discriminant analysis (LDA) score > 2.0, $p < 0.05$). 22 and 8 of these genera remained significantly different after adjusting for the age and gender using multivariate association with linear models (MaAsLin2) method (Fig. 3a, b). The increased abundance of genera *Streptococcus*, *Fusobacterium*, and *Gemella*, was observed in oCRC group, and the genera *Fusobacterium*, *Flavonifractor* and *Odoribacter*, contained the key phylotypes in the yCRC group. We further identified genera *Faecalibacterium* and *Blautia* as key microbiota in the oControl and yControl group, respectively (Fig. 3a, b). The statistical distribution and relative abundance of these specific genus-level biomarkers support the prevalence of the bacteria in most of the samples.

Given gut microbial dysbiosis have been associated with tumor stage and tumor location[20,21], we first catalog the microbiome signature across different disease stages in oCRC and yCRC. The genera *Fusobacterium* and *Christensenellaceae_R7* were enriched in stage 0–III and stage IV oCRC, respectively, while genus *Faecalibacterium* was the significantly distinct bacteria dominant in age-matched healthy control (Supplementary Fig. 1a). In yCRC, genera Erysipelotrichaceae_UCG003, *UCG005*, and *Faecalibacterium* were the dominant microbiota in the stage 0–III

young-onset patients, stage IV young-onset patients, and age-matched healthy control, respectively (Supplementary Fig. 1b). We further compared the changes in the microbiome signature of oCRC and yCRC according to tumor location. As shown in Supplementary Fig. 1c, the increased abundance of genera *Parvimonas* and *Fusobacterium*, were observed in old-onset left-side colon cancer (LCC) and right-side colon cancer (RCC), respectively. By contrast, the butyrate-producing bacteria *Faecalibacterium* contained the main phylotype in the age-matched healthy controls for the oCRC. In the comparison of microbial differences among young people, we identified genera *Alistipes* and *Roseburia* as key microbiota in the young-onset LCC and RCC, respectively, which was accompanied by significant increase of *Escherichia Shigella* in age-matched healthy control (Supplementary Fig. 1d). These results indicate significant interactions between gut microbiota and clinicopathological characteristics.

To further obtain deeper insights into gut microbiota species identification and metabolic imputation, metagenomic sequencing with species-level taxonomic resolution was performed using samples selected randomly from Fudan cohort ($n = 50$ per group). The strong associations of community composition

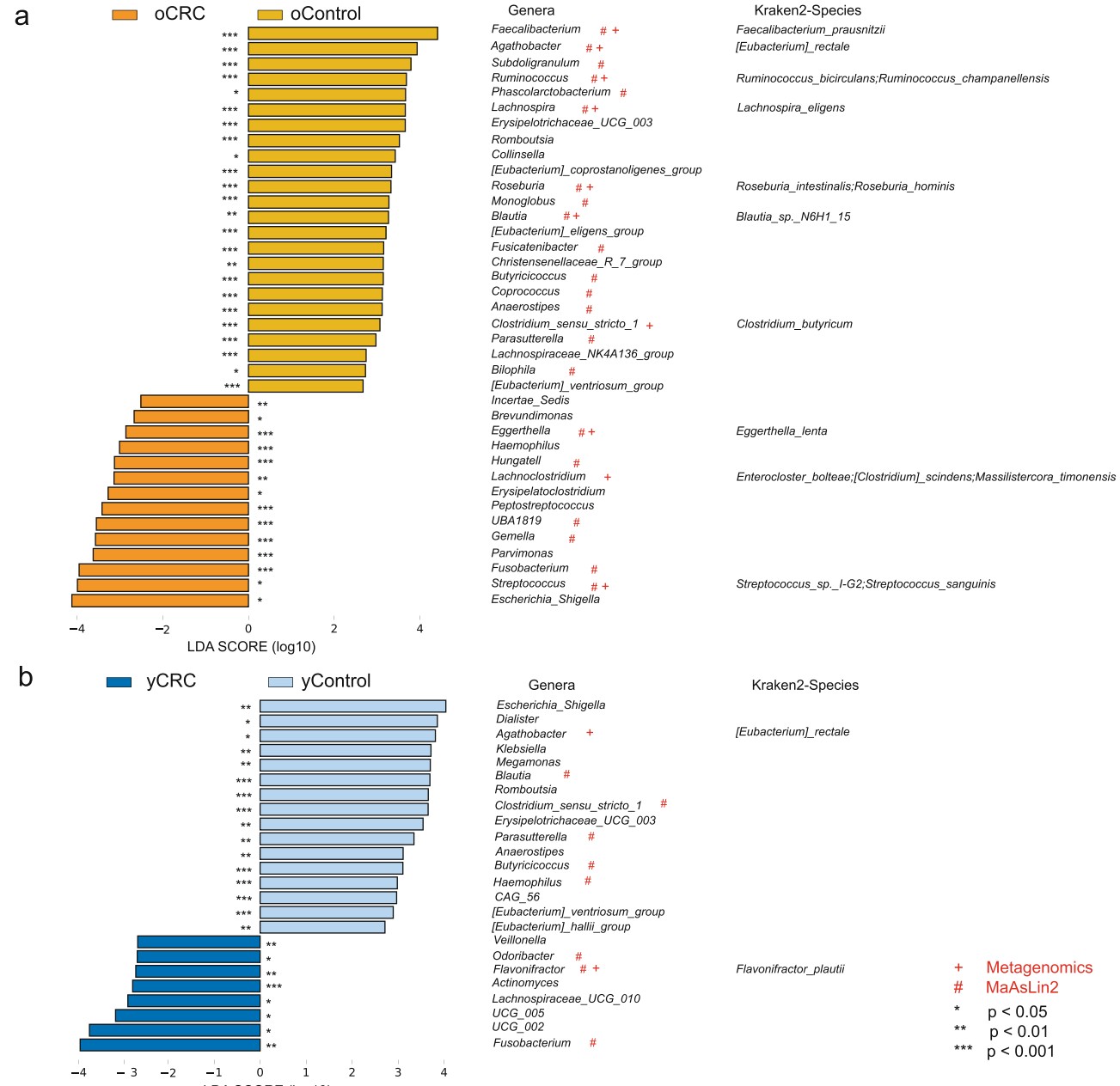

**Fig. 3 Phylogenetic profiles of fecal microbial communities in yCRC and oCRC. a** Histogram of LDA coupled with effective size measurement based on the 16S rRNA gene sequencing (adjusting by MaAsLin2, $n = 233$ for oCRC and $n = 203$ for oControl), and the metagenomic sequencing ($n = 50$ in each group) between oCRC and oControl. $p$ values are calculated by Kruskal–Wallis test, logarithmic LDA score > 2.0, $p < 0.05$. **b** LEfSe based on the 16S rRNA gene sequencing (adjusting by MaAsLin2, $n = 144$ for yCRC and $n = 148$ for yControl), and the metagenomic sequencing ($n = 50$ in each group) between yCRC and yControl. $p$ values are calculated by Kruskal–Wallis test, LDA score > 2.0, $p < 0.05$. '#' showed bacterial taxa with distinct relative abundances between groups detected by 16S rRNA gene sequencing after adjusting for the age and gender using MaAsLin2; '+' suggested the significantly differential bacteria between groups detected by metagenomic sequencing. LDA linear discriminant analysis, CRC colorectal cancer, yCRC young-onset CRC, oCRC old-onset CRC, yControl age-matched healthy controls for the yCRC, oControl age-matched healthy controls for the oCRC. Source data are provided as a Source Data file.

between 16S rRNA sequencing and metagenomic sequencing was confirmed by Procrustes analysis (Protest was performed to test for significance. oCRC versus oControl: $M^2 = 0.7934$, $p = 0.004$; yCRC versus yControl: $M^2 = 0.9265$, $p = 0.043$) and Mantel test (oCRC versus oControl: $r = 0.1936$, $p = 0.011$; yCRC versus yControl: $r = 0.1961$, $p = 0.021$) (Supplementary Fig. 2a, b). Compared to 16S rRNA sequencing, metagenomic sequencing supported that genus *Streptococcus* was still the key microbiota in the oCRC group, whereas genus *Flavonifractor* was the dominant

microbiota in the yCRC group (Supplementary Fig. 2c, d). Then, species-level microbial markers were also characterized between CRC patients (oCRC, yCRC) and age-matched healthy controls (oControl, yControl) by LEfSe (Supplementary Fig. 2c, d). Compared to yControl, bacterial species of *Flavonifractor plautii*, which was reported affecting antigen-induced T helper 2 cell (Th2) immune responses in mice[22], was consistently the dominant population in yCRC detected by both 16S rRNA and metagenomic sequencing. *Enterocloster bolteae*, *Lachnospire*

*eligens*, and *Eggerthella lenta* contained the key phylotypes in the oCRC compared to oControl. Correspondingly, butyric acid-producing bacteria *Faecalibacterium prausnitzii* and *Eubacterium rectale* were enriched in oControl and yControl, respectively[23]. The statistical distribution and relative abundance of these species-level biomarkers showed the prevalence of them in most of the samples. Obviously, the correlation between metagenomics and 16S rRNA gene sequencing data highlights the robustness of our results.

**Functional analysis of fecal microbiota**. It was reported that microbiota imbalance can induce systematic metabolic alterations[24,25], while metabolic dysfunction can in turn influence microbiota composition[26]. To study the functional and metabolic changes of the fecal microbial communities, all the clean reads from metagenomic sequencing were aligned to the suggested database to obtain Kyoto Encyclopedia of Genes and Genomes (KEGG) modules and Gene Ontology (GO) enrichment from bacterial species (Supplementary Fig. 3). As a result, the KEGG modules involved in short-chain fatty acid metabolism were overrepresented in health (7 from 25 modules in oControl, 3 from 8 modules in yControl) compared to CRC (2 from 19 modules in oCRC, 1 from 5 modules in yCRC) (Supplementary Fig. 3a, b). In GO-enrichment analysis, the plasma membrane and protein binding were the key terms increased in oCRC, while translation, transposase activity, and structural constituent of ribosome were overrepresented in oControl (Supplementary Fig. 3c). Notably, the GO enrichments between yCRC and yControl displayed sparse differential terms, characterized by the dominant of DNA binding and RNA-dependent DNA biosynthetic process in yCRC, indicated stronger cell proliferation and invasion ability (Supplementary Fig. 3d). Altogether, our results suggested that young-onset or oCRC patients have their unique bacterial metabolic features.

**Identification and validation of fecal microbial ASVs-based markers for oCRC and yCRC**. To evaluate the classification power of fecal bacteria markers for oCRC and yCRC, a random forest classifier (RFC) model was constructed. We performed a tenfold cross-validation on a random forest model in the training phase (70% of the samples randomly selected from the Fudan cohort including 163 oCRC and 142 oControl) to detect unique ASVs-based markers for oCRC. Our analysis identified the top 40 differentially abundant markers as the optimal marker set between 163 oCRC and 142 oControl (Fig. 4a, b). The probability of disease (POD) index was then calculated using the identified 40 ASVs-based markers for both the training group and the testing group. In the training phase, the average POD value was significantly increased in the oCRC group versus the oControl ($p = 2.22e{-}16$, Fig. 4c), and the POD index achieved an area under receiving operating characteristics curves (AUC) value of 89.28% (95% CI: 85.84–92.72%) (Fig. 4d). In the testing phase (the remaining 30% samples from Fudan cohort, including 70 oCRC and 61 oControl), the average POD value increased significantly in oCRC group as compared with the oControl group ($p = 2.4e{-}12$, Fig. 4e), and the AUC value of the microbial markers was 86.67% (95% CI: 80.31–93.04%) (Fig. 4f).

Similarly, the strength of fecal microbiome in distinguishing yCRC from age-matched control was also illustrated by constructing a RFC model. In the training phase (70% samples randomly selected from Fudan cohort, including 100 yCRC and 104 yControl), top 60 differentially abundant genera markers were selected as the optimal marker set (Fig. 5a, b). The POD value was significantly increased in the yCRC group vs. the yControl ($p = 2.22e{-}16$, Fig. 5c), and the POD index achieved an AUC

value of 86.57% (95% CI: 81.78–91.36%) (Fig. 5d). In the testing phase (the remaining 30% samples from Fudan cohort, including 44 yCRC and 44 yControl), the average POD value was significantly increased ($p = 7e{-}06$) in yCRC group, and the POD achieved an AUC value of 79.52% (95% CI: 69.85–89.2%) (Fig. 5e, f).

To measure the strength of observed associations, the 146 oCRC, 41 yCRC, 54 oControl, and 69 yControl from Huadong cohort were served as independent external validation set (Supplementary data 2). The results showed that the average POD value in the oCRC was significantly higher than that of oControl ($p = 1.8e{-}15$, Fig. 6a), and the POD achieved an AUC value of 86.67% (95% CI: 79.31–94.03%) (Fig. 6b). Also, the average POD value was significantly elevated in the yCRC versus yControl ($p = 5.1e{-}10$), and the POD achieved an AUC value of 87.8% (95% CI: 80.82–94.78%) (Fig. 6c, d). Collectively, our results show that POD based on fecal microbial markers has a strong power for distinguishing yCRC or oCRC from health.

**Predicting performance of fecal microbial markers for yCRC and oCRC**. Clinically, both non-quantitative FOBT or quantitative fecal immunochemical test (FIT) are non-invasive methods for CRC screening[27]. CEA and CA19-9 are the two most widely used serum tumor markers, especially in diagnosis and prognosis of advanced CRC[28]. In order to compare the RFC microbial markers with the conventional screening methods, we used two independent group, including 728 subjects in the Fudan cohort and 310 subjects in the Huadong cohort. As a result, using non-quantitative FOBT, CEA, or CA19-9 levels as a predictor alone between 233 oCRC cases and 203 oControl in the Fudan cohort generated an AUC of 0.5833, 0.7439, and 0.6691, respectively; however, the combination of CEA and CA19-9 achieved an AUC of 0.8199 (Fig. 6e). Simultaneously, an AUC of 0.5566, 0.7711, 0.6984, and 0.8254, respectively, were obtained between 144 yCRC and 148 yControl (Fig. 6f). Notably, the RFC model we constructed could discriminate the samples of yCRC or oCRC from respective age-matched controls with an AUC of 0.8981 or 0.8511, respectively, which significantly improve predictive performance (Fig. 6e, f). Furthermore, the strength of microbial model for distinguishing yCRC or oCRC from health was validated by comparing with non-quantitative FOBT, CEA, and CA19-9 in Huadong cohort. The non-quantitative FOBT, CEA, CA19-9 alone, and the combined CEA and CA19-9 yielded an AUC of 0.5076, 0.7435, 0.6721 and 0.7851 to discriminate 146 oCRC from 54 oControl, while the microbial markers increased the AUC to 0.8656 (Fig. 6g). Similar results were found between 41 yCRC and 69 yControl in Huadong cohort when compared non-quantitative FOBT, CEA, CA19-9 alone, or the combined CEA and CA19-9 (AUC: 0.6203, 0.7296, 0.5832, and 0.7252, respectively) with the microbial markers (AUC: 0.8561) (Fig. 6h). In general, our results suggest the great potential of using gut microbiota biomarkers as a promising non-invasive tool in detection and distinction of yCRC and oCRC. Patients with yCRC and oCRC have their unique fecal microbial markers for better distinguishing them from health.

**Discussion**
In CRC, it is still challenging to early evaluate the yCRC that has increasing aggressiveness and unclear underlying mechanisms[29]. In recent years, the microbiome signature has attracted extensive attention in various disease fields due to its excellent performance in early diagnosis and prognosis evaluation[30–32]. Meanwhile, increasing studies have shown that microbiome signatures are characterized by age sequence and colorectal adenoma-carcinoma sequence[33,34]. Given these evidences, we strive to explore the intestinal microbial composition and function of yCRC, and

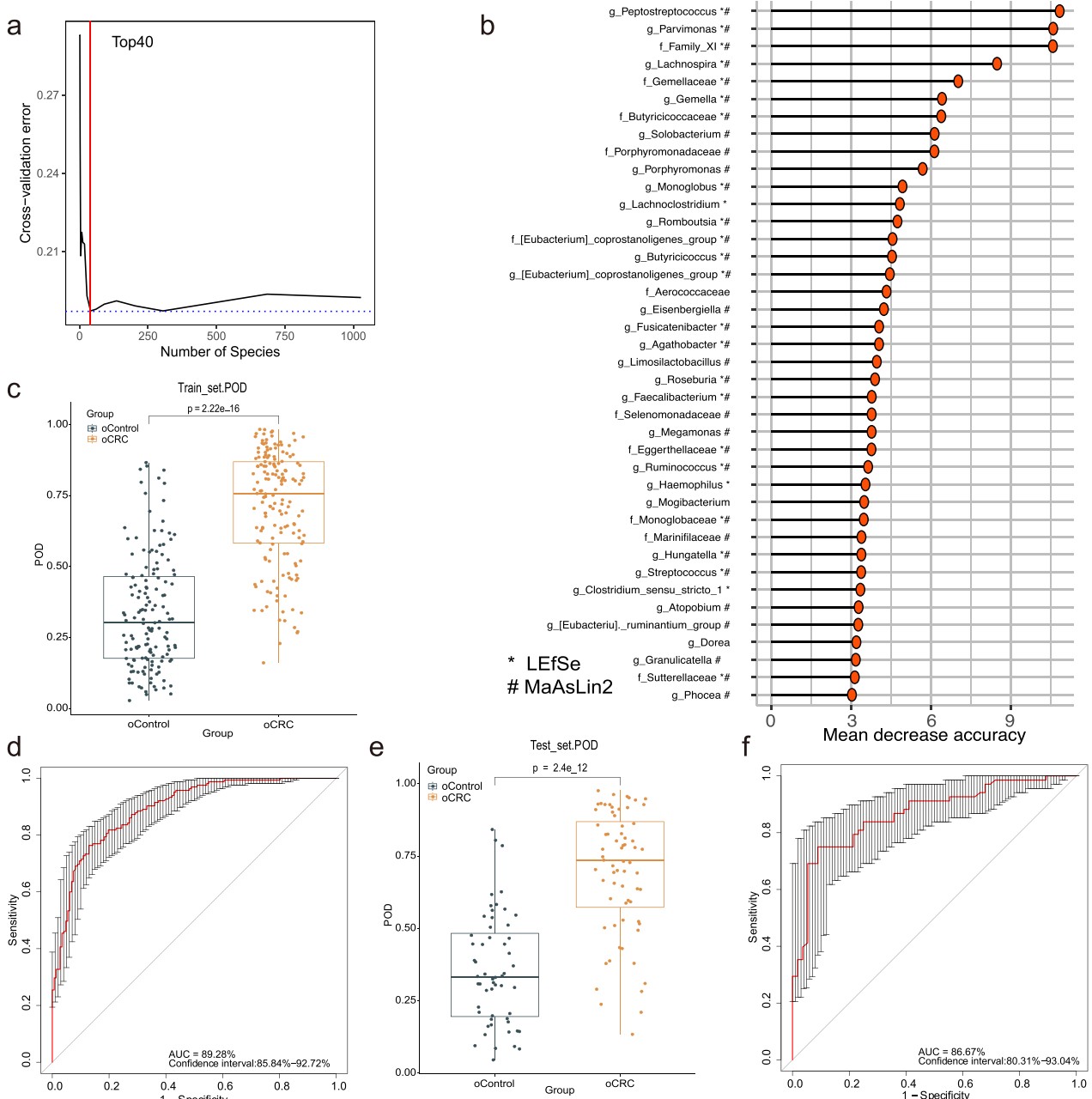

**Fig. 4 Identification of microbial markers of oCRC by random forest models. a** In the training phase, a tenfold cross-validation was performed on a random forest model between 163 oCRC cases and 142 oControl. **b** Top 40 differentially abundant markers were selected as the optimal marker set based on random forest between 163 oCRC cases and 142 oControl. The x-axis presents the mean decrease accuracy to each marker, which indicates the contribution to the accuracy of the model; '#' showed bacterial taxa with distinct relative abundances between groups detected by 16S rRNA gene sequencing after adjusting for the age and gender using MaAsLin2; '*' suggested the relative abundance differences derived from LEfSe analysis of 16S rRNA gene sequencing. **c** The POD value in oCRC ($n = 163$) vs. oControl ($n = 142$) in the training set. oCRC vs. oControl, The box denotes 25th– 75th percentiles and the central mark indicates the median; the whiskers are 1.5 times the interquartile range; dots outside the whiskers indicate outliers; p value is calculated by two-sided unpaired Mann–Whitney test. **d** The POD-based AUC value between oCRC ($n = 163$) and oControl ($n = 142$) in the training set. Error bars denote 95% confidence interval for AUC value. **e** The average POD value in 70 oCRC vs. 61 oControl in the testing phase. oCRC vs. oControl. The box denotes 25th–75th percentiles and the central mark indicates the median; the whiskers are 1.5 times the interquartile range; dots outside the whiskers indicate outliers; p value is calculated by two-sided unpaired Mann–Whitney test. **f** The POD-based AUC value between oCRC ($n = 70$) and oControl ($n = 61$) in the testing phase. Error bars denote 95% confidence interval for AUC value. AUC area under the curve, CRC colorectal cancer, oCRC old-onset CRC, oControl age-matched healthy controls for the oCRC, POD probability of disease, SD standard deviation. Source data are provided as a Source Data file.

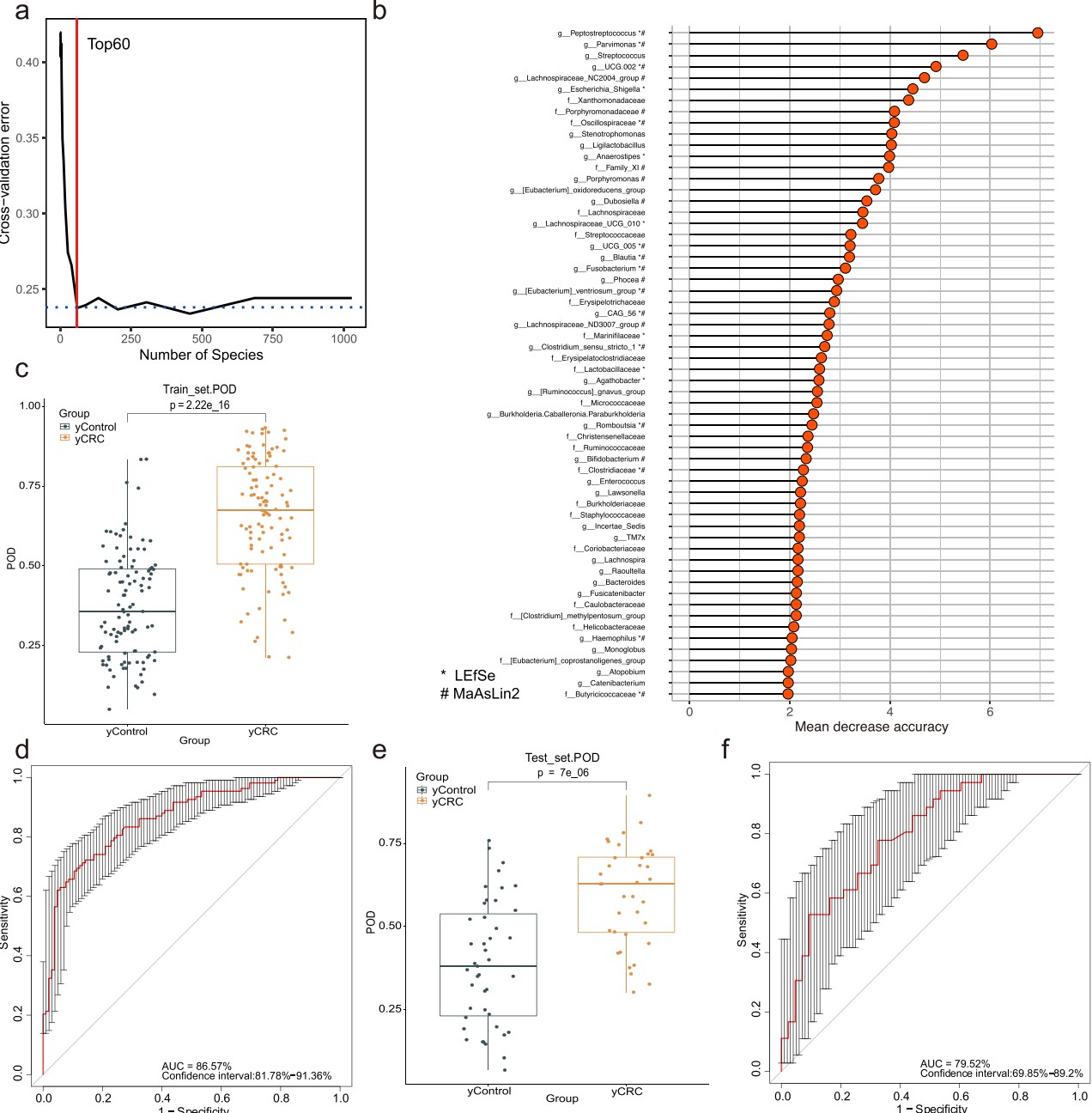

**Fig. 5 Identification of microbial markers of yCRC using random forest models. a** A tenfold cross-validation on a random forest model between 100 yCRC and 104 yControl in the training set. **b** Top 60 differentially abundant genera markers were selected as the optimal marker set based on random forest between 100 yCRC and 104 yControl. The x-axis presents the mean decrease accuracy to each marker, which indicates the contribution to the accuracy of the model; '#' showed bacterial taxa with distinct relative abundances between groups detected by 16S rRNA gene sequencing after adjusting for the age and gender using MaAsLin2; '*' suggested the relative abundance differences derived from LEfSe analysis of 16S rRNA gene sequencing. **c** The POD value in yCRC (n = 100) versus yControls (n = 104) in the training set. yCRC vs. yControl. The box denotes 25–75th percentiles and the central mark indicates the median; the whiskers are 1.5 times the interquartile range; dots outside the whiskers indicate outliers; p value is calculated by two-sided unpaired Mann–Whitney test. **d** The POD-based AUC value between yCRC (n = 100) and yControl (n = 104) in the training set. Error bars denote 95% confidence interval for AUC value. **e** The average POD value in 44 yCRC vs. 44 yControl in the testing phase. yCRC vs. yControl. The box denotes 25th–75th percentiles and the central mark indicates the median; the whiskers are 1.5 times the interquartile range; dots outside the whiskers indicate outliers; p value is calculated by two-sided unpaired Mann–Whitney test. **f** The POD-based AUC value between yCRC (n = 44) and yControl (n = 44) in the testing phase. Error bars denote 95% confidence interval for AUC value. AUC area under the curve, CRC colorectal cancer, yCRC young-onset CRC, yControl age-matched healthy controls for the yCRC, POD probability of disease. Source data are provided as a Source Data file.

establish the microbiome signature for better identifying yCRC from age-matched healthy people and distinguishing them from elderly patients. Our efforts resulted in the discovery of unique microbiome signatures for the oCRC (n = 233) and yCRC

(n = 144) versus age-matched control (oControl and yControl), thereby successfully discriminating them from training and testing cohorts. Importantly, microbial markers successfully achieved an independent validation of oCRC (n = 146) and yCRC (n = 41)

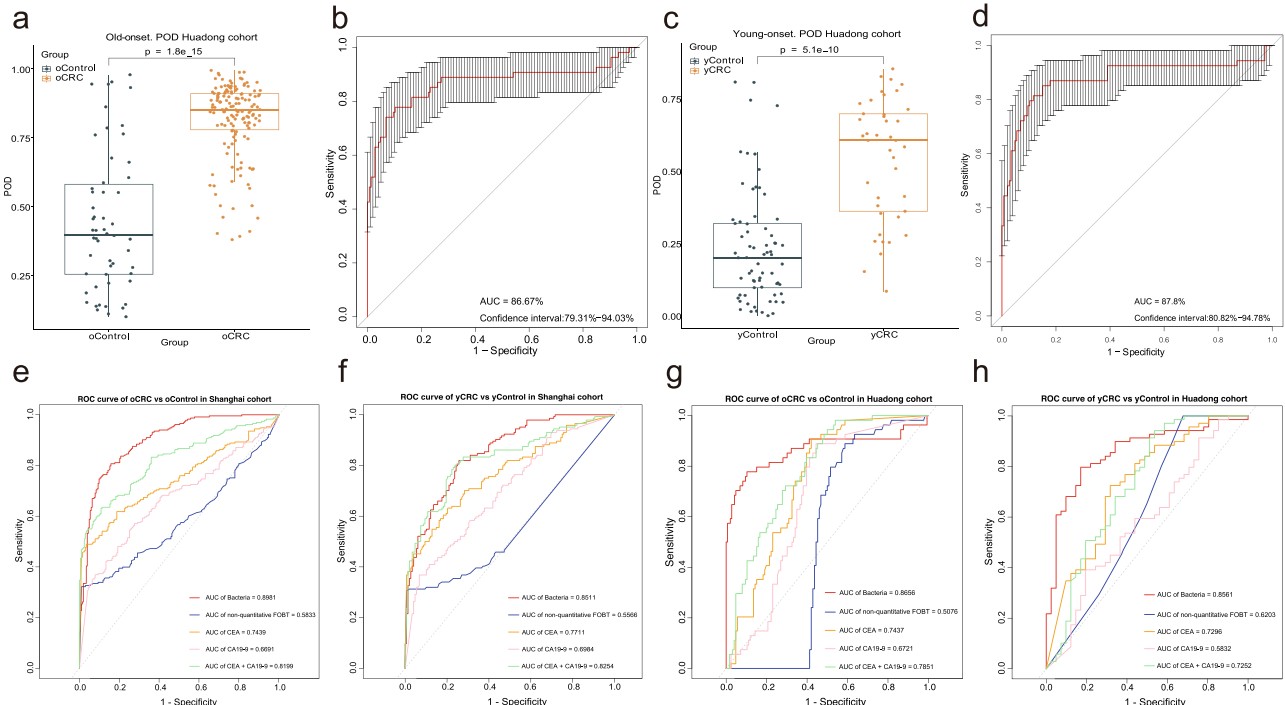

**Fig. 6 Independent validation and diagnostic performance of fecal microbial markers for yCRC and oCRC. a** The POD value compared between 146 oCRC and 54 oControl in the independent external validation phase from Huadong cohort. oCRC vs. oControl. The box denotes 25th–75th percentiles and the central mark indicates the median; the whiskers are 1.5 times the interquartile range; dots outside the whiskers indicate outliers; *p* value is calculated by two-sided unpaired Mann–Whitney test. **b** The POD-based AUC value between oCRC (*n* = 146) and oControl (*n* = 54) in the Huadong cohort. Error bars denote 95% confidence interval for AUC value. **c** The POD value compared between 41 yCRC and 69 yControl in the independent external validation phase from Huadong cohort. yCRC vs. yControl. The box denotes 25th–75th percentiles and the central mark indicates the median; the whiskers are 1.5 times the interquartile range; dots outside the whiskers indicate outliers; *p* value is calculated by two-sided unpaired Mann–Whitney test. **d** The POD-based AUC value between yCRC (*n* = 41) and yControl (*n* = 69) in the Huadong cohort. Error bars denote 95% confidence interval for AUC value. **e** and **f** AUC values for the prediction of oCRC (**e**) and yCRC (**f**) using microbial markers, non-quantitative FOBT, serum CEA, serum CA19-9, or the combined CEA and CA19-9 among 203 oControl, 233 oCRC, 148 yControl, 144 yCRC in Fudan cohort. **g** and **h** AUC values for the prediction of oCRC (**g**) and yCRC (**h**) using microbial markers, non-quantitative FOBT, serum CEA, serum CA19-9, or the combined CEA and CA19-9 among 54 oControl, 146 oCRC, 69 yControl, 41 yCRC in Huadong cohort. ROC, receiver operating characteristic curves; AUC area under the curve, CRC colorectal cancer, yCRC young-onset CRC, oCRC old-onset CRC, yControl age-matched healthy controls for the yCRC, oControl age-matched healthy controls for the oCRC, FOBT fecal occult blood test, CEA carcinoembryonic antigen, CA19-9 carbohydrate antigen 19-9. Source data are provided as a Source Data file.

from Huadong cohort. Furthermore, microbial markers could help identify CRC in the young people and CRC in the elderly. These findings collectively indicated that the fecal microbiome would be a promising non-invasive approach for early screening of yCRC and oCRC, respectively.

Prior studies have reported the association of microbial dysbiosis with senescence and chronic disease[35]. Firstly, we analyzed the diversity of the fecal microbiome among the groups and found that the diversity of the oCRC group was lower, while the fecal microbiota of patients with yCRC exhibited higher diversity than that of oCRC. As biological age increases, overall gut microbiota richness decreases, while some microbial taxa associated with aging emerge. Moreover, high bacterial diversity often occurs in a variety of diseases and is considered to be a type of microbial dysbiosis, which challenges the general assumption that reduced alpha-diversity is usually associated with CRC microbiota dysbiosis[36], suggesting the particularity of gut microbes in young-onset patients. We also compared the differential enriched fecal microbiota among groups, and noted that the *Streptococcus*, *Fusobacterium*, and *Gemella*, whose connection with CRC has been widely reported[37–39], had significantly increased abundance in oCRC patients. We identified the yCRC-associated genera including *Fusobacterium* and *Flavonifractor*. Moreover, bacterial species of *Flavonifractor plautii*, a flavonoid-degrading bacterium,

was consistently the dominant population in yCRC detected by both sequencing methods in our study. *Fusobacterium*, especially *Fusobacterium nucleatum*, is an opportunistic pathogen of many chronic oral and intestinal diseases, and the infection rate increases with age[40]. Unexpectedly, our study identified *Fusobacterium* as an important microbiota for CRC in young and old people. It cannot be explained by the fact that pathogenic microorganisms can easily induce inflammation and immunodeficiency in elderly patients[41]. However, this result implies that *Fusobacterium* has a strong mucosal adhesion ability in the general population. *Flavonifractor*, a rare clinical pathogens that may induce oxidative stress and systemic inflammation in alcoholic hepatitis patients[42], has not been found to be associated with aging. It is noteworthy that a recent study identified *Flavonifractor plautii* as one of the key bacterium associated with CRC in the Indian races[43]. Considering that flavonoids are a large number of polyphenolic compounds in plant-based diets[44], our above findings reveal that the incidence of CRC in young people may be more closely related to diet and lifestyle.

Then, we conducted function analysis and found that the DNA binding and RNA-dependent DNA biosynthetic process pathway were overrepresented in yCRC group, indicated stronger cell proliferation and invasion ability. A previous study demonstrated that tumor cells alter their metabolism to meet their demand for

macromolecules and support a high rate of proliferation as well as respond to oxidative stress[45]. Among the metabolic reprogramming in tumor cell, pentose phosphate pathway was stimulated and the production of nucleotides and DNA synthesis was increased, which in turn reduced intracellular reactive oxygen species levels, and promoted antioxidant defense[45]. Therefore, the yCRC-specific microbiota manifested as a metabolic state that is more prone to malignant progression, supporting the poorer prognosis of young-onset patients.

Last but not least, we identified specific fecal microbial markers for distinguishing yCRC or oCRC from age-matched healthy controls, and validated the strength of observed associations based on random forest classification models. As the results showed, classifier based on optimal 40 differentially abundant genus-level markers achieved a high accuracy between oCRC and oControl in training cohort, which also achieved a high accuracy between oCRC group and healthy controls in testing cohorts and independent validation cohort. Similarly, the classifier based on optimal 60 differentially abundant genus-level markers for distinguishing yCRC from yControl also achieved a good performance in training cohort, testing cohort and independent validation cohort. These findings suggested that fecal microbiota-based biomarkers were potentially helpful in predicting the risks of oCRC and yCRC, as well as distinguishing CRC onset based on age. A recent practice guideline recommended reducing the age to initiate CRC screening to 45 years old[46]. However, the cost-effectiveness analysis shows that lowering the age of screening also brought many socioeconomic problems[47]. Thus, better risk assessment and personalized screening strategies can help improve the detection of at-risk populations. Given the accuracy for non-invasively detecting yCRC of our study, it may help raise awareness about CRC in the young, and promote colonoscopy screening in certain young populations with CRC-related microbiome signatures to reduce the incidence of sporadic CRC.

In addition to the fecal microbiota that has received increasing attention, FOBT, including either guaiac-based (gFOBT, non-quantitative FOBT) or immunological-based (iFOBT, FIT), is also a non-invasive method and is currently implemented for CRC screening in average-risk population[48–50]. Specifically, non-quantitative FOBT detects hemoglobin, while quantitative FIT detects globin and is not affected by diet[51]. The diagnostic accuracy of non-quantitative FOBT in the published literature ranged from 25.5% to 86.0% with sensitivity and specificity values ranged from 7.4–75.0% and 21.6–98.6%, respectively[52–56]. FIT for hemoglobin shows better CRC diagnostic performance than non-quantitative FOBT[57]. However, its high diagnostic sensitivity comes at the cost of a high false positive rate[58,59]. Furthermore, the optimal threshold of FIT for CRC screening is still unknown because the fecal hemoglobin concentration varies with gender and age[60]. A systematic review with meta-analysis reported that FIT had a pooled sensitivity of 91% and a pooled specificity of 90% in detecting CRC using a positivity threshold of 10 µg/g, whereas a threshold ≥ 20 µg/g resulted in a pooled sensitivity of 75% and a pooled specificity of 0.95[61]. This review is highly heterogeneous by population setting, age range, FIT brand and FIT threshold used. In a primary care population with low-risk symptoms of CRC, the AUC for the FIT was 0.92[62]. Another review further quantified the performance characteristics of FIT for CRC stratified by age, with a pooled sensitivity of 85% for ages 50–59 and a sensitivity of 73% for ages 60–69, but did not compare the diagnostic efficacy of FIT in young people under 50[63]. In our study, the sensitivity and specificity for the detection of CRC with the gut microbiota biomarkers were comparable with non-quantitative FOBT but inferior to FIT data published in the above-mentioned literature. However, our study not only emphasizes the use of gut microbiota biomarkers as a promising

non-invasive tool to detect CRC, but we also highlights its use to distinguish yCRC and oCRC. The potential precision screening effect of fecal microbiota on yCRC provide a unique opportunity for clinical practice, which may help early identification of more individuals with yCRC.

Strengths of our study include multi-center study design, utilization of both metagenomic and 16S rRNA gene sequencing for microbial phylogeny and functional analysis, comparison with other commonly used non-invasive testing methods, as well as the age-matched assessment. However, whether our results are consistent between regions with different races still needs to be verified by other multi-center studies with larger number of subjects.

In conclusions, our current study reveals the common state of fecal microbial dysbiosis in yCRC and oCRC populations. Fecal microbiota-based biomarkers have a robust strength in distinguishing yCRC from aged-matched control. Although more clinical validations and mechanism investigations are needed, our study emphasizes the need to further study the potential association between the gut microbiota and the risk of CRC in young people, which may drive the clinical transformation of microbiota-based strategies into precision screening and diagnosis.

## Methods

**Characteristics of the participants and sample collection.** In total, 1071 fecal samples were collected from year 2018 to 2021 in Fudan University Shanghai Cancer Center, Shanghai, China (Fudan cohort), Tongji University Affiliated Tenth People's Hospital, Shanghai (Huadong cohort), China and The Second Hospital of Shandong University, Shandong, China (Huadong cohort). After inclusion and exclusion screenings, a total of 1038 eligible subjects were included in our study, with 728 participants from Fudan cohort and 310 participants from Huadong cohort. Eligible subjects were randomly divided into training phase, testing phase, and validation phase (Fig. 1). Each fecal sample was collected in a sterile tube and then stored at −80 °C prior to microbial analysis. None of the participants were treated with antibiotics or probiotics one month before enrollment in this study. For sporadic CRC group, fecal samples were collected preoperatively and participants were excluded based on the following criteria: the history of familial CRC, the history of inflammation-associated CRC, the history of irritable bowel syndrome (IBS), with other coexisting malignant tumors, stool sampling not before colonoscopy or neoadjuvant therapy before stool sampling. Recruited CRC patients were divided into two groups according to age: yCRC group, age <50 years old; oCRC group, age ≥50 years old. For healthy control group, volunteers confirmed as no gastrointestinal tumors after colonoscopy screening were recruited and were divided into two groups based on age: yControl group, age <50 years old; oControl group, age ≥50 years old. The clinical pathological features of CRC including age, gender, tumor location, tumor size, tumor differentiation, TNM stage, KRAS/NRAS/BRAF gene mutation status, non-quantitative FOBT, CEA, CA19-9, and lymphatic/nerve/vascular invasion status were recorded. A chemiluminescent microparticle immunoassay method was performed to detect CEA and CA19-9 in preoperative blood samples. Cutoff values recommended for diagnostic purposes were 37 kU/L for CA19-9 and 5.9 mg/L for CEA. Values above the cutoff concentrations were considered positive. A dual-qualitative FOBT method was used for detecting hemoglobin and transferrin in fecal samples before colonoscopy. Ethical approval was obtained from the Institutional Review Board of Fudan University Shanghai Cancer Center, and written informed consent was provided by all subjects before sampling.

**Fecal DNA extraction for microbiome analysis.** Genomic DNA of fecal samples was extracted using the QIAamp DNA Stool Mini Kit (Qiagen, Hilden, Germany) according to the manufacturer's guidelines. DNA integrity and size were verified by 1.0% agarose gel electrophoresis and DNA concentrations were determined using NanoDrop spectrophotometry (Nano Drop, Germany).

**High-throughput 16S ribosomal RNA gene sequencing.** 16S ribosomal RNA (rRNA) gene amplification was performed using the primers (319F: 5′-ACTCC-TACGGGAGGCAGCAG-3′; 806R: 5′-GGACTACHVGGGTWTCTAAT-3′) directionally targeting the V3 and V4 hypervariable region of the 16S rRNA gene. To differentiate each sample and yield accurate phylogenetic and taxonomic information, the gene products were attached with forward and reverse error-correcting barcodes. The amplicons were quantified after purification. Then, the normalized equimolar concentrations of each amplicon were pooled and sequenced on the MiSeq PE300 sequencing instrument (Illumina) using 2 × 300 bp chemistry according to the manufacturer's specifications.

**DNA Library construction and metagenomic sequencing**. Sequencing libraries were constructed by TruSeq Nano DNA LT Library Preparation Kit (Illumina). DNA was fragmented by dsDNA Fragmentase (NEB) and incubated at 37 °C for 30 min. Library construction began with fragmented cDNA. Blunt-end DNA fragments were generated by fill-in reactions and exonuclease activity. Provided sample purification beads were used for size selection. An A-base was then added to the blunt ends of each strand, preparing them for ligation to the indexed adapters. Each adapter contained a T-base overhang for ligating the adapter to the A-tailed fragmented DNA. These adapters contained the full complement of sequencing primer hybridization sites for single, paired-end, and indexed reads. Single- or dual-index adapters were ligated to the fragments and the ligated products were amplified with PCR. After libraries purification, quantification and quality control, high-throughput sequencing was carried out on the NovaSeq6000 platform (Illumina) according to the manufacturer's specifications.

**Sequencing data analysis**

*As for 16S rRNA gene sequencing data*. The raw data of 16S rRNA gene sequencing were analyzed using QIIME2 platform (v2020.2). In briefly, DADA2 plugin was used to filter the sequencing reads and to construct ASVs feature table. The taxonomy information of ASVs were assigned against the Silva Database (https://www.arb-silva.de) (v138.1) (download code: qiime2 rescript get-silva-data–p-version '138.1'–p-target 'SSURef_NR99') using classify-sklearn algorithm by feature-classifier plugin. A phylogenetic tree was generated using FastTree [QIIME2 platform (v2020.2)] and Mafft [QIIME2 platform (v2020.2)] alignment by phylogeny plugin. Alpha and beta diversity analyses were conducted using diversity plugin. Bacterial diversity was presented by observed count of ASVs and Shannon index. PCoA was conducted to display distance among samples. To assess the effects of different phenotypes on gene profiles among groups, PERMANOVA was performed.

*As for metagenomic sequencing data*. Raw sequencing reads were processed to obtain valid reads for further analysis. First, sequencing adapters were removed from sequencing reads using cutadapt (v1.9). Secondly, low-quality reads were trimmed by fqtrim (v0.94) using a sliding-window algorithm. Thirdly, reads were aligned to the host genome using bowtie2 (v2.2.0) to remove host contamination. Once quality-filtered reads were obtained, they were de novo assembled to construct the metagenome for each sample by IDBA-UD (v1.1.1). All coding regions (CDS) of metagenomic contigs were predicted by MetaGeneMark (v3.26). CDS sequences of all samples were clustered by CD-HIT (v4.6.1) to obtain unigenes. Unigene abundance for a certain sample were estimated by transcripts per kilobase million (TPM) based on the number of aligned reads by bowtie2 (v2.2.0). The lowest common ancestor taxonomy of unigenes were obtained by aligning them against the NCBI NR database by DIAMOND (v0.9.14). Similarly, the functional annotation (GO [http://geneontology.org/] and KEGG [https://www.kegg.jp/]) of unigenes were obtained. In order to obtain the species-level information, the clean reads were aligned to the suggested database (v202003) using Kraken2 software (v2.1.1) and Braken software (v2.5). The database can be freely download from the Kraken2 website (https://ccb.jhu.edu/software/kraken2/index.shtml?t=downloads).

**Biomarker identification and POD construction**. Based on ASVs frequency profile, the frequency profile at family-level and genus-level were selected for further analysis. A five trials of tenfold cross-validation was conducted to identify optimal microbial biomarkers using a machine learning method (random forest, RF), and the cut-off point was selected by the mean of minimum cross-validation error. The top most discriminatory biomarkers were selected by mean decrease accuracy (a feature importance score in random forest model), and they were considered as the optimal set with minimum error. The POD index was calculated according to the article published previously[64]. In brief, POD index was considered as the ratio that a sample could be predicted as CRC and that of healthy controls from the number of randomly generated decision trees in random forest model. The identified optimal set of biomarkers was finally used for the calculation of POD index for both the training cohort (70% of the Fudan cohort), testing cohort (30% of the Fudan cohort) and independent cohort (Huadong cohort). Receiver operating curve (ROC) and area under curve (AUC) were used to evaluate the strength of the constructed models.

**Statistical analyses**. Comparison between quantitative data was conducted using the unpaired Student's *t*-test, Mann–Whitney *U*-test, or Dunnett's *t*-test, where appropriate. The associations between the clinical characteristics were performed by Pearson's Chi-square test or Fisher's exact test. Linear discriminant analysis effect size (LEfSe, https://huttenhower.sph.harvard.edu/galaxy/) was used to identify taxonomic and functional features which are differentially abundant between cases and controls. To evaluate and deconfound the effects of age and gender, multivariate association with linear models algorithm (MaAsLin2, http://huttenhower.sph.harvard.edu/maaslin) was used for multivariable association testing between phenotypes and microbial taxonomy or functional characters with default parameters. A tenfold cross-validation was conducted on a random forest model to identify optimal microbial markers. Receiver operating characteristic (ROC) curve was used to evaluate the performance of multivariable that differentiate between certain groups. Spearman correlation analysis was performed to analyze the correlation between the gut microbiota and functional characters. Procrustes analysis and Mantel test were used to evaluate the association between 16S data and metagenome data. All *P* values were two-tailed and *P* values of 0.05 or less were considered to be statistically significant. All data were analyzed by the Graph Pad Prism 8.0 software (Graph Pad software, lnc., San Diego, CA, USA), R version 3.6.3 (R Foundation for Statistical Computing, Vienna, Austria, http://www.R-project.org/) and Microsoft Excel (Microsoft Corporation, Seattle, WA, USA).

**Reporting summary**. Further information on research design is available in the Nature Research Reporting Summary linked to this article.

## Data availability
16S rRNA gene sequences data and the metagenomic sequences data that support the findings of this study are publicly available at the NIH National Center for Biotechnology Information Sequence Read Archive (SRA) with BioProject ID PRJNA763023. Individual accession codes are provided in Supplementary Data 3. Used databases are Silva Database (https://www.arb-silva.de) (v138.1) (download code: qiime2 rescript get-silva-data–p-version '138.1'–p-target 'SSURef_NR99'), NCBI NR database (https://www.ncbi.nlm.nih.gov/), GO (http://geneontology.org/), and KEGG (https://www.kegg.jp/). The remaining data are available within the Article, Supplementary Information, or Source Data file. Source data are provided with this paper.

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

## Acknowledgements

The authors take this opportunity to thank all of the participating patients and healthy volunteers for supporting this study by donating the precious samples used in this research. This work was supported by grants from the National Natural Science Foundation of China (Nos. 81920108026 and 81871964 for Y.M.; 81802412 for Y.Y.), the National Ten Thousand Plan Young Top Talents (for Y.M.), the Shanghai Young Top Talents (No. QNBJ1701 for for Y.M.), the Shanghai Science and Technology Development Fund (No. 19410713300 for for Y.M.), the Shanghai Sailing Program (No. 18YF1414900 for Y.Y.), the CSCO-Roche Tumor Research Fund (No. Y-2019Roche-079 for for Y.M.) and the Fudan University Excellence 2025 Talent Cultivation Plan (for Y.M. and Y.Y.).

## Author contributions

Y.Y., D.S., C.K. and Y.M. designed the experiments. Y.Y., L.D., D.S., J.L., C.K., X.L. and Y.M. provided the clinical samples and performed the experiments. G.L., Y.Y., C.K. and Y.M. analyzed the data. G.L., Y.Y., C.K. and Y.M. wrote the manuscript. All authors edited the manuscript.

## Competing interests

The authors declare no competing interests.
