## [Peer Review File · Nature Communications]

Reviewers' Comments:

Reviewer #1:

Remarks to the Author:

This is an interesting study comparing the faecal microbiome of 144 young-onset CRC, 233 old-onset CRC and 233 healthy volunteers (patient's spouse or partner). The study found specific faecal microbiome signature that is unique to young-onset CRC and old-onset CRC which can reasonably distinguish CRC and healthy controls.

Comments:

- Given microbiome signature have been associated with age as well as tumor location (right vs left sided colorectal cancer has been shown to have different microbiome signature), the study should aim to have age-matched controls and also compare microbiome signature according to tumor location between the yCRC and oCRC.
- Was there any difference in microbiome signature by stage (early vs stage IV) and recurrence or survival outcome?
- The authors mentioned that faecal microbiome could be used as a diagnostic/screening test for colorectal cancer. It would be useful if the diagnostic accuracy of the microbial signature can be compared to the faecal occult blood test which is the current screening test. Serum marker CEA or CA19-9 though can be prognostic, they are not currently used for cancer screening and does not add value to this study.

Reviewer #2:

Remarks to the Author:

The authors are addressing an important need in the field of colorectal cancer by turning to the fecal microbiome as a potential biomarker to aid in the diagnosis of young-onset colorectal cancer.

The cohort assembled and the analyses presented in the report provide a great stepping stone to build on top. Before moving forward with publication - please address the following concerns:

Major Comments:

- Biomarker discovery is impacted by the quality of data used as the input. Authors should employ stringent denoising and filtering of the 16S dataset to reduce the impact of potential contaminants and artifacts. 16S methods are prone to producing noise and using 16S data processing tools on default mode is not enough to produce clean datasets. The estimated alpha diversity values and indices are extremely high - not unusual to see this from mothur/qiime when run with default parameters. The beta-diversity results also require attention - a significant p-value is reported but there is not a great deal of variability in this dataset according to the PCs plotted and there seems to be a great deal of overlap from the PCs plotted as well. In addition, there seems to be variability not explained by group status (N, yCRC, oCRC).
- OTU generation and taxonomic classification provide a good way to start the analysis but this approach does not offer enough resolution to build robust biomarkers. I strongly suggest the evaluation of this dataset using species-level taxonomic resolution offered by metagenomic sequencing which will lead to better biomarker identification and metabolic imputation. Other 16S approaches may also offer better denoising and improved resolution like DADA2.
- The authors didn't provide any framework that addresses potential confounders or effect modifiers. It is important to include this in the analysis as a number of known covariates in cancer patients can act as confounders. Table 1 already offers information about potential confounders that need to be adjusted for.
- Metabolic imputation is a hypothesis-generating exercise and these results should not be considered as final.
- As seen in other indications, these results might be cohort-specific. The use of a validation cohort drawn from the same population limits the robustness of the classifier. Given the small number of

OTUs shared between yCRC and N and an even smaller number of OTUs specific to yCRC, it is important to validate this signal with additional cohorts.

Minor comments:

- check spelling and grammar
- improve the quality of the figures submitted
- remove redundancy in LEfSe plots. Plots include OTUs that map to the same lineage and same values appear to be reported at different taxonomic levels.

Reviewer #3:

Remarks to the Author:

The current manuscript by Yongzhi Yang and colleagues used a case-control approach for comparative analyses of gut microbiota to identify young CRC patients from healthy people and explore the differences between younger and elderly CRC patients. The authors have performed a detailed analysis for identifying the microbiome signatures in patient of young-onset CRC. The major strengths of this paper are the use of well-established sequencing technique and a large sample size to analyze the microbiota. Furthermore, the figures are well prepared and the results supports well the conclusions of the paper. All together, this is a manuscript of interest for the large spectrum of readers of the journal. However, the manuscript will greatly benefit if the authors address the following concerns.

Comment:

1. The introduction section of the manuscript (Line 63, page 4) mentions that the incidence of CRC under 55 years old is approximately 10%. In the "Experimental section" (Line 328, page 15), the cut-off value of young CRC patients and old CRC patients is set to 50. The author needs to explain why the cutoff value is 50 instead of 55.
2. In the "Experimental section" (Line 321, page 15), please explain when, where, and how the primary fecal specimens were handled for collected and saved (Line 322, page 15).
3. In the "Experimental section", the author do not describe the use of probiotics, which has influence on the structure of the gut microbiome. Please improve the inclusion and exclusion criteria in the section of "Characteristics of the Participants and Sample Collection"(line 321-336, page 15).
4. The control group in Table 1 shows that there are 3 normal controls with high CEA and 2 normal controls with high CA19-9 (Line 104-107, page 5-6, and Table 1). Please explain whether these participants have further examinations to exclude gastrointestinal tumors.
5. For Figure 2F, this manuscript mentioned that it was plotted using unweighted UniFrac distance method (Line 124-127, page 6 and line 560-562, page 24). The PCoA algorithm should be used instead of PCA (PCA plot is a special case of PCoA plot by using Euclidean distance). The authors need to clarify this issue.
6. For Figure 3A, the authors mentioned linear discriminant analysis coupled with effect size analysis (line 130-131, page 7), but the analysis method is not described in the material method. Explain why.
7. For Figure 4A, the paper described it was PCA plot (line 160, page 8). However, the caption of Figure 4 indicated that it was PCoA plot (line 572, page 25). Is the mention of " PCA" in error?
8. For Figure 5, The probability of disease (POD) index was calculated (Line 179, page 9), please provide reference for the POD method and described POD algorithm in the material method.
9. For Figure 7, In discussing the diagnostic value, the paper found fecal microbial OTU-based markers improve predictive performance compared with single tumor maker CEA or CA19-9. It is recommended to add following point: compare the efficacy of the combined detection of CEA and CA199 markers with that of the combined detection of fecal microbial OTU-based markers. Please assess the diagnostic value both the discovery set and the validation set.
10. The font size at all levels in each picture is not uniform. Please keep the font size of titles uniform in each figure.
11. The acronym needs to be displayed in its full form when it first appears, such as LDA (Line 564, page 24). Please check the consistency throughout the whole manuscript.
12. Some references in the Introduction section of the manuscript are out of date, please refer to the current studies.
13. There are several incomplete sentences and syntax, grammatical errors in the manuscript. Few

are listed below, please correct:

- a. In the Figure Legends, "156 patients 546 with yCRC and 241 patients with oCRC..."(line 545-546, page 24). Here, the sentence is incorrect, Please correct the sentence to make it easier for the reader to understand. Please check the consistency throughout the whole manuscript.
- b. Discussion and conclusion section (line 306, page 14) "10 OTUs markers achieved ..." should be "10 OTU markers achieved ...". Similar inconsistencies also appeared in Supplementary Table 4 and Table 5 (line 716 and 719, page 37).
- c. Statistical Analyses section (Line 379, page 17) mentioned "P or FDR values less than 0.05 were designated as significant differences (*, < 0.05, **, < 0.01, ***, < 0.001)". The * that appears may be redundant.
- d. Discussion and conclusion section (line 306, page 14) mentioned "which warrants following mechanism investigations in vivo and vitro". "in vivo and vitro" should be "in vivo and in vitro" and in italics.

Response Letter

Review #1

1. Comment: Given microbiome signature have been associated with age as well as tumor location (right vs left sided colorectal cancer has been shown to have different microbiome signature), the study should aim to have age-matched controls and also compare microbiome signature according to tumor location between the yCRC and oCRC.

Answer: Thank you very much for your valuable suggestion. We have reanalyzed our data using age-matched healthy controls. We have modified our abstract, results, discussion, and figure accordingly (color coded).

we further compared the changes in the microbiome signature of oCRC_and yCRC according to tumor location. As shown in **Figure S1C**, the increased abundance of genus *Parvimonas* and *Fusobacterium*, were observed in old-onset left-side colon cancer (LCC) and right-side colon cancer (RCC), respectively. By contrast, the butyrate-producing bacteria *Faecalibacterium* contained the main phylotype in the old-onset healthy control. In the comparison of microbial differences among young people, we identified genus *Alistipes* and *Roseburia* as key microbiota in the young-onset LCC and RCC, respectively, which was accompanied by significant increase of *Escherichia_Shigella* in age-matched healthy control (**Figure S1D**). These results indicate significant interactions between gut microbiota and clinicopathological characteristics.

.According to your suggestion, we have supplemented these results (color coded), and Figure S1C-D, in the revised version of our manuscript.

Figure S1C. Histogram of linear discriminant analysis coupled with effective size measurement based on the 16S rRNA gene sequencing between 203 old healthy control and 233 old-onset CRC in different tumor locations (left-side colon and right-side colon). LDA score >2.0 , $p < 0.05$. LCC, left-side colon cancer; RCC, right-side colon cancer.

Figure S1D. Histogram of linear discriminant analysis coupled with effective size measurement based on the 16S rRNA gene sequencing between 148 young healthy control and 144 young-onset CRC in different tumor locations (left-side colon and right-side colon). LDA score >2.0 , $p < 0.05$. LCC, left-side colon cancer; RCC, right-side colon cancer.

2. Comment: Was there any difference in microbiome signature by stage (early vs stage IV) and recurrence or survival outcome?

Answer: We greatly appreciate your comments. Unfortunately, the fecal samples from CRC were collected during 2018 to 2021, resulting in incomplete data for recurrence or survival outcome. So, these contents were not included in this study. We will make efforts to clarify it in our future work.

According to your advice, we compared microbiome signature among tumor stages (TNM stage I-III, IV) in yCRC (**Figure S1A**) and oCRC (**Figure S1B**). The genus *Fusobacterium* and *Christensenellaceae_R7* were enriched in stage 0-III and stage IV

old-onset CRC, respectively, while genus *Faecalibacterium* was the significantly distinct bacteria dominant in age-matched healthy control (**Figure S1A**). In young-onset CRC, genus *Erysipelotrichaceae_UCG003*, *UCG005*, and *Faecalibacterium* were the dominant microbiota in the stage 0-III young-onset patients, stage IV young-onset patients, and age-matched healthy control, respectively (**Figure S1B**). We have supplemented these results (color coded), and **Figure S1A-B**, in the revised version of our manuscript.

Figure S1A. Histogram of linear discriminant analysis coupled with effective size measurement based

on the 16S rRNA gene sequencing between 203 old healthy control and 233 old-onset CRC in different tumor stages (TNM stage 0-III and IV).

Figure S1B. Histogram of linear discriminant analysis coupled with effective size measurement based on the 16S rRNA gene sequencing between 148 young healthy control and 144 young-onset CRC in different tumor stages (TNM stage 0-III and IV).

3. Comment: The authors mentioned that faecal microbiome could be used as a diagnostic/screening test for colorectal cancer. It would be useful if the diagnostic accuracy of the microbial signature can be compared to the faecal occult blood test which is the current screening test. Serum marker CEA or CA19-9 can be prognostic, they are not currently used for cancer screening and does not add value to this study.

Answer: Thank you for this excellent suggestion. We have assessed the diagnostic value of microbial markers by comparing with fecal immunochemical tests.

As a result, using FIT, CEA, or CA19-9 levels as a predictor alone between 233 oCRC cases and 203 oControl in the Fudan cohort generated an AUC of 0.5833, 0.7439, and 0.6691 respectively; however, the combination of CEA and CA19-9 achieved an AUC of 0.8199 (**Figure 7E**). Simultaneously, an AUC of 0.5566, 0.7711, 0.6984, and 0.8254 respectively were obtained between 144 yCRC and 148 yControl (**Figure 7F**). Notably, the random forest classifier model we constructed could discriminate the samples of yCRC or oCRC from respective age-matched controls with an AUC of 0.8981 or 0.8511, respectively, which significantly improve predictive performance (**Figure 7E, F**). Furthermore, the diagnostic efficacy of the microbial model was validated by comparing with FIT, CEA, and CA19-9 in Huadong cohort. The FIT, CEA, CA19-9 alone, and the combined CEA and CA19-9 yielded an AUC of 0.5076, 0.7435, 0.6721 and 0.7851 to discriminate 146 oCRC from 54 oControl, while the microbial markers increased the AUC to 0.8656 (**Figure 7G**). Similar results were found between 41 yCRC and 69 yControl in Huadong cohort when compared FIT, CEA, CA19-9 alone, or the combined CEA and CA19-9 (AUC: 0.6203, 0.7296, 0.5832, and 0.7252, respectively) with the microbial markers (AUC: 0.8561) (**Figure 7H**). In general, our results suggest the potential of using faecal microbiome to develop a specific diagnostic test for yCRC or oCRC. yCRC and oCRC groups have their unique fecal microbial markers for better diagnostic accuracy.

We have modified the results (color coded) and **Figure 7** accordingly.

Figure 7

Figure 7. Independent validation and diagnostic performance of fecal microbial markers for yCRC and oCRC.

Reviewer #2

1. Comment: Biomarker discovery is impacted by the quality of data used as the input. Authors should employ stringent denoising and filtering of the 16S dataset to reduce the impact of potential contaminants and artifacts. 16S methods are prone to producing noise and using 16S data processing tools on default mode is not enough to produce clean datasets. The estimated alpha diversity values and indices are extremely high - not unusual to see this from mothur/qiime when run with default parameters. The beta-diversity results also require attention - a significant p-value is reported but there is not a great deal of variability in this dataset according to the PCs plotted and there seems to be a great deal of overlap from the PCs plotted as well. In addition, there seems to be variability not explained by group status (N, yCRC, oCRC).

Answer: Thank you very much for your valuable suggestions. Firstly, according your advice, we have denoised our 16S rRNA gene data as follows: ‘The raw data of 16S rRNA gene sequencing were analyzed using QIIME2 platform (v2020.2). In briefly, DADA2 plugin was used to filter the sequencing reads and construct ASVs (Amplicon Sequence Variants) feature table, and the taxonomy information of ASVs was assigned against the Silva Database (v138.1) using classify-sklearn algorithm by feature-classifier plugin. A phylogenetic tree was generated using FastTree and Mafft alignment by phylogeny plugin. Alpha and beta diversity analyses were conducted using diversity plugin. Principal coordinates analysis (PCoA) was conducted to display distance between samples.’.

After denoising using DADA2, we obtained an average of 28455 16S rRNA gene sequences per sample (min: 4794; max: 143517; median:23660), and 40032 Amplicon Sequence Variants (ASVs). Then, we reevaluated the differences of bacterial diversity in four groups (old-onset CRC, old control, young-onset CRC, young control), sequences

were aligned for alpha-diversity. Updated indices for alpha-diversity can be found in **Table S3**.

We also recalculated the beta diversity using weighted UniFrac method, and performed principal coordinates analysis (PCoA) to display microbiome space between samples in the four groups. The results presented a significantly separated distribution among groups using permutational multivariate analysis of variance analysis (**Figure 2D**). We have modified the results (color coded) and **Figure 2** accordingly.

Figure 2

Figure 2. Bacterial diversity of the fecal microbiota associated with yCRC and oCRC. (A-B) Fecal microbial diversity was estimated by the Chao1 index (A) and Shannon index (B). (C) A Venn diagram displays the overlaps between groups. (D) Beta diversity was calculated using weighted UniFrac by PCoA. The microbiome distribution between samples was calculated using PERMANOVA. oControl, old healthy controls; oCRC, old-onset colorectal cancer; yControl, young

healthy controls; yCRC, young-onset colorectal cancer. PCoA, principal coordinate analysis. PERMANOVA, permutational multivariate analysis of variance analysis

2. Comment: OTU generation and taxonomic classification provide a good way to start the analysis but this approach does not offer enough resolution to build robust biomarkers. I strongly suggest the evaluation of this dataset using species-level taxonomic resolution offered by metagenomic sequencing which will lead to better biomarker identification and metabolic imputation. Other 16S approaches may also offer better denoising and improved resolution like DADA2.

Answer: We very much appreciate these critical comments. As our response in your first comment, we have denoised our 16S rRNA gene data by using QIIME2 platform and DADA2 plugin. To further obtain deeper insights into gut microbiota species identification and metabolic imputation, metagenomic sequencing with species-level taxonomic resolution was also performed between old-onset CRC, old control, young-onset CRC, young control (n = 50 per group). We have modified the methods, results (color coded), and **Figure 2-7, Figure S2** accordingly.

3. Comment: The authors didn't provide any framework that addresses potential confounders or effect modifiers. It is important to include this in the analysis as a number of known covariates in cancer patients can act as confounders. Table 1 already offers information about potential confounders that need to be adjusted for.

Answer: Thank you for this useful suggestion. We used MaAsLin2 (Multivariate Association with Linear Models algorithm, <http://huttenhower.sph.harvard.edu/maaslin>) for multivariable association between phenotypes and microbial taxonomy or functional characters with default parameters, to evaluate the effects of age and gender. We have revised the methods, results (color coded), and **Figure 3,5-6, Figure S2** according to your suggestion.

4. Comment: Metabolic imputation is a hypothesis-generating exercise and these results should not be considered as final.

Answer: Thanks for your insightful suggestion. To gain insights into gut microbiota species identification and metabolic imputation, metagenomic sequencing with species-level taxonomic resolution was also performed between old-onset CRC, old control, young-onset CRC, young control (n = 50 per group) in the revised version of our manuscript. All the clean reads from metagenomic sequencing were aligned to the suggested database using Kraken2 software and Braken software to obtain Kyoto Encyclopedia of Genes and Genomes (KEGG) modules and Gene Ontology (GO) enrichment from bacterial species. Based on metagenomic sequencing and deeper modules database, our results displayed more reliable bacterial metabolic features. According to your suggestion, we have modified the methods, results (color coded), and **Figure 4**, in the revised version of our manuscript.

Figure 4

Figure 4. Fecal microbial functional dysbiosis in yCRC and oCRC. (A-B) Fecal microbial functions between oControl and oCRC (A), and between yControl and yCRC (B) were predicted based on Kyoto Encyclopedia of Genes and Genomes (KEGG) modules. (C-D) Fecal microbial functions between oControl and oCRC (C), and between yControl and yCRC (D) were predicted based on Genomes Gene ontology (GO). oControl, old healthy controls; oCRC, old-onset colorectal cancer; yControl, young healthy controls; yCRC, young-onset colorectal cancer.

5. Comment: As seen in other indications, these results might be cohort-specific. The use of a validation cohort drawn from the same population limits the robustness of the classifier. Given the small number of OTUs shared between yCRC and N and an even smaller number of OTUs specific to yCRC, it is important to validate this signal with additional cohorts.

Answer: Thank you for this excellent suggestion. Firstly, instead of OTUs, family-level and genus-level amplicon sequence variants (ASVs) frequency profile was selected for further analysis. A tenfold cross-validation was conducted to identify optimal microbial biomarkers using random forest, and the cut-off point was selected by the mean of minimum cross-validation error. As a result, top 40 differentially abundant genus markers were selected as the optimal marker set for old-onset CRC (**Figure 5**), while top 60 differentially abundant genus markers for young-onset CRC (**Figure 6**).

To further confirm the diagnosis potential of random forest model, a new cohort including 146 oCRC, 41 yCRC, 54 oControl, and 69 yControl from Huadong China (Huadong cohort) were served as independent diagnostics to verify the POD reliability. The results showed that average POD value was significantly higher in the oCRC than that in oControl ($p < 0.0001$, **Figure 7A**), and the AUC value achieved 86.67% (95% CI: 79.31%-94.03%) (**Figure 7B**). Also, the average POD value was significantly elevated in the yCRC versus yControl ($p < 0.0001$), and the POD achieved an AUC value of 85.56% (95% CI: 78.38%-92.74%) (**Figure 7C, D**). These results suggested that POD based on

microbial markers has a strong diagnostic efficacy for oCRC and yCRC patients from Huadong China. We have revised the manuscript (color coded) and **Figure 5-7** accordingly.

6. Comment: Check spelling and grammar.

Answer: The modifications have been made to the revised manuscript. In addition, we have obtained the services of a professional editing company to improve the writing in our revised manuscript.

7. Comment: Improve the quality of the figures submitted.

Answer: The quality of the figures have been improved in the revised version of our manuscript. Thanks.

8. Comment: Remove redundancy in LEfSe plots. Plots include OTUs that map to the same lineage and same values appear to be reported at different taxonomic levels.

Answer: Thank you for pointing out the mistake. We've reanalyzed the data and avoided the same problem in the new results.

Reviewer #3

The current manuscript by Yongzhi Yang and colleagues used a case-control approach for comparative analyses of gut microbiota to identify young CRC patients from healthy people and explore the differences between younger and elderly CRC patients. The authors have performed a detailed analysis for identifying the microbiome signatures in patient of young-onset CRC.

The major strengths of this paper are the use of well-established sequencing technique and a large sample size to analyze the microbiota. Furthermore, the figures are well prepared and the results supports well the conclusions of the paper. All together, this is a manuscript of interest for the large spectrum of readers of the journal. However, the manuscript will greatly benefit if the authors address the following concerns.

Comment:

1. The introduction section of the manuscript (Line 63, page 4) mentions that the incidence of CRC under 55 years old is approximately 10%. In the “Experimental section” (Line 328, page 15), the cut-off value of young CRC patients and old CRC patients is set to 50. The author needs to explain why the cutoff value is 50 instead of 55.

Answer: We thank the reviewer for this valuable suggestion. Few studies have tried to systematically define “young-onset.” The definition of young-onset CRC varies across published studies, where patients are frequently designated under arbitrary age cutoffs ranging anywhere between 40 and 60 years¹. One of the objective of our study was to understand how to more effectively screen and diagnosis for CRC in patients currently below the current recommended screening ages. Most guidelines set the starting age for CRC screening at 50 years old². Thus, the cut-off value of young CRC patients and old CRC patients is set to 50 in our study. In order to reduce the confusion of the readers of this article, we have revised the description of the age of young-onset CRC in the section of “Introduction”.

References:

1. Jacobs D, Zhu R, Luo J, et al. Defining Early-Onset Colon and Rectal Cancers. *Front Oncol* 2018;8:504.

2. Helsingen LM, Vandvik PO, Jodal HC, et al. Colorectal cancer screening with faecal immunochemical testing, sigmoidoscopy or colonoscopy: a clinical practice guideline. *BMJ* 2019;367:l5515.

2. In the “Experimental section” (Line 321, page 15), please explain when, where, and how the primary fecal specimens were handled for collected and saved (Line 322, page 15).

Answer: Thanks for your insightful suggestion. Each fecal sample was collected in a sterile tube and then stored at -80°C prior to microbial analysis. None of the participants were treated with antibiotics or probiotics one month before enrollment in this study. In addition, for sporadic CRC group, cohort, fecal samples were collected preoperatively. According to your suggestion, we have modified the section of “MATERIALS AND METHODS” in the revised version of our manuscript.

3. In the “Experimental section”, the author do not describe the use of probiotics, which has influence on the structure of the gut microbiome. Please improve the inclusion and exclusion criteria in the section of “Characteristics of the Participants and Sample Collection”(line 321-336, page 15).

Answer: Thanks for your insightful suggestion. None of the participants were treated with antibiotics or probiotics one month before enrollment in this study. We have modified the section of “MATERIALS AND METHODS” in the revised version of our manuscript.

4. The control group in Table 1 shows that there are 3 normal controls with high CEA and

2 normal controls with high CA19-9 (Line 104-107, page 5-6, and Table 1). Please explain whether these participants have further examinations to exclude gastrointestinal tumors.

Answer: Thank you for your detailed interpretation of this article. We have integrated the requirements of all reviewers and added control samples and a validation set sample. After inclusion and exclusion screening, a total of 1038 eligible subjects were included in our study, with 728 participants from Fudan cohort and 310 participants from Huadong cohort. For healthy control group. Volunteers confirmed as no gastrointestinal tumors after colonoscopy screening were recruited. Therefore, all healthy control samples have been excluded from gastrointestinal tumor diseases through gastrointestinal endoscopy. The relevant description has been modified in the revised version of our manuscript.

5. For Figure 2F, this manuscript mentioned that it was plotted using unweighted UniFrac distance method (Line 124-127, page 6 and line 560-562, page 24). The PCoA algorithm should be used instead of PCA (PCA plot is a special case of PCoA plot by using Euclidean distance). The authors need to clarify this issue.

Answer: We thank the reviewer for this critical comment. We recalculated the beta diversity using weighted UniFrac method, and performed principal coordinates analysis (PCoA) to display microbiome space between samples in the four groups. The results presented a significantly separated distribution among groups using permutational multivariate analysis of variance analysis (**Figure 2D**). We have modified the results and Figure accordingly.

Figure 2D Beta diversity was calculated using weighted UniFrac by PCoA. The microbiome distribution between samples was calculated using PERMANOVA.

6. For Figure 3A, the authors mentioned linear discriminant analysis coupled with effect size analysis (line 130-131, page 7), but the analysis method is not described in the material method. Explain why.

Answer: We appreciated that reviewer help us to find this mistake. Linear discriminant analysis effect size (LEfSe, <https://huttenhower.sph.harvard.edu/galaxy/>) was used to identify taxonomic and functional features which are differentially abundant between cases and controls. We have added the description of the analysis method about LEfSe in the revised manuscript.

7. For Figure 4A, the paper described it was PCA plot (line 160, page 8). However, the caption of Figure 4 indicated that it was PCoA plot (line 572, page 25). Is the mention of "PCA" in error?

Answer: We appreciated that reviewer propose this suggestion. In our original manuscript, we used PICRUSt to perform a functional analysis of the metabolic function of the flora, and based on the prediction results of the PICRUSts function, we used STAMP and PCA plot for difference analysis. PICRUSt conducts research on community

system evolution through recessive state reconstruction. The software is based on 16S rDNA and reference sequence database to predict the functional composition of metagenomics. However, compared with 16s rRNA gene sequencing, metagenomic sequencing has a richer reflection on the comparison of species and metabolic functions.

To gain insights into gut microbiota species identification and metabolic imputation, metagenomic sequencing with species-level taxonomic resolution was performed between old-onset CRC, old control, young-onset CRC, young control (n = 50 per group) in the revised version of our manuscript. All the clean reads from metagenomic sequencing were aligned to the suggested database using Kraken2 software and Braken software to obtain Kyoto Encyclopedia of Genes and Genomes (KEGG) modules and Gene Ontology (GO) enrichment from bacterial species. Based on metagenomic sequencing and deeper modules database, our results displayed more reliable bacterial metabolic features. We have modified the results in the revised version of our manuscript.

Figure 4

Figure 4. Fecal microbial functional dysbiosis in yCRC and oCRC. (A-B) Fecal microbial functions between oControl and oCRC (A), and between yControl and yCRC (B) were predicted based on Kyoto Encyclopedia of Genes and Genomes (KEGG) modules. (C-D) Fecal microbial functions between oControl and oCRC (C), and between yControl and yCRC (D) were predicted based on Genomes Gene ontology (GO). oControl, old healthy controls; oCRC, old-onset colorectal cancer; yControl, young healthy controls; yCRC, young-onset colorectal cancer.

8. For Figure 5, The probability of disease (POD) index was calculated (Line 179, page 9), please provide reference for the POD method and described POD algorithm in the material method.

Answer: We thank the reviewer for this valuable suggestion. We have cited the following reference in the section of MATERIALS AND METHODS of the revised manuscript: Ren Z, Li A, Jiang J, et al. Gut microbiome analysis as a tool towards targeted non-invasive biomarkers for early hepatocellular carcinoma. Gut 2019;68:1014-1023.

Based on ASVs frequency profile, family-level and genus-level frequency profile was selected for further analysis. A tenfold cross-validation was conducted to identify optimal microbial biomarkers using a machine learning method (random forest, RF), and the cut-off point was selected by the mean of minimum cross-validation error. The top most discriminatory biomarkers were selected by feature importance in random forest model, and they were considered as the optimal set with minimum error. The POD index was calculated according to the article published previously. In briefly, POD index was considered as the ratio that a sample could be predicted as “Colorectal cancer” and that of healthy controls from the number of randomly generated decision trees in random forest model. The identified optimal set of biomarkers was finally used for the calculation of POD index for both the training cohort (70% of the Fudan cohort), testing cohort (30% of the Fudan cohort) and independent cohort (Huadong cohort). Receiver operating curve (ROC) and area under curve (AUC) were used to evaluate the diagnostic performance of the constructed models.

We have added the description of POD algorithm in the section of MATERIALS AND METHODS of the revised manuscript.

9. For Figure 7, In discussing the diagnostic value, the paper found fecal microbial OTU-based markers improve predictive performance compared with single tumor marker CEA or CA19-9. It is recommended to add following point: compare the efficacy of the combined detection of CEA and CA199 markers with that of the combined detection of fecal microbial OTU-based markers. Please assess the diagnostic value both the discovery set and the validation set.

Answer: We appreciated that reviewer propose this suggestion. In order to compare the diagnostic efficacy of microbial markers with the conventional non-invasive and invasive clinical diagnostic methods, we used two independent group, including 728 subjects in the Fudan cohort and 310 subjects in the Huadong cohort. As a result, using FIT, CEA, or CA19-9 levels as a predictor alone between 233 oCRC cases and 203 oControl in the Fudan cohort generated an AUC of 0.5833, 0.7439, and 0.6691 respectively; however, the combination of CEA and CA19-9 achieved an AUC of 0.8199 (**Figure 7E**). Simultaneously, an AUC of 0.5566, 0.7711, 0.6984, and 0.8254 respectively were obtained between 144 yCRC and 148 yControl (**Figure 7F**). Notably, the random forest classifier model we constructed could discriminate the samples of yCRC or oCRC from respective age-matched controls with an AUC of 0.8981 or 0.8511, respectively, which significantly improve predictive performance (**Figure 7E, F**). Furthermore, the diagnostic efficacy of the microbial model was validated by comparing with FIT, CEA, and CA19-9 in Huadong cohort. The FIT, CEA, CA19-9 alone, and the combined CEA and CA19-9 yielded an AUC of 0.5076, 0.7435, 0.6721 and 0.7851 to discriminate 146 oCRC from 54 oControl, while the microbial markers increased the AUC to 0.8656 (**Figure 7G**). Similar results were found between 41 yCRC and 69 yControl in Huadong cohort when compared FIT, CEA, CA19-9 alone, or the combined CEA and CA19-9 (AUC: 0.6203, 0.7296, 0.5832, and 0.7252, respectively) with the microbial markers (AUC: 0.8561) (**Figure 7H**). In general, our results suggest the potential of using faecal microbiome to develop a specific diagnostic test for yCRC or oCRC. yCRC and oCRC groups have their unique fecal microbial markers for better diagnostic accuracy.

Figure 7

Figure 7. Independent validation and diagnostic performance of fecal microbial markers for yCRC and oCRC.

10. The font size at all levels in each picture is not uniform. Please keep the font size of titles uniform in each figure.

Answer: Thank you for this good suggestion. We have revised the figures of our manuscript accordingly.

11. The acronym needs to be displayed in its full form when it first appears, such as LDA (Line 564, page 24). Please check the consistency throughout the whole manuscript.

Answer: Thanks to the reviewer for the question. We have revised accordingly and check the consistency throughout the whole manuscript.

12. Some references in the Introduction section of the manuscript are out of date, please refer to the current studies.

Answer: We appreciate that reviewer suggest us to update the reference. We have revised accordingly.

13. There are several incomplete sentences and syntax, grammatical errors in the manuscript. Few are listed below, please correct:

Answer: We appreciated that you help us to find these mistakes. We have gone through our manuscript and checked for the grammar, style structure and punctuation.

a. In the Figure Legends, “156 patients 546 with yCRC and 241 patients with oCRC...”(line 545-546, page 24). Here, the sentence is incorrect, Please correct the sentence to make it easier for the reader to understand. Please check the consistency throughout the whole manuscript.

Answer: We thank the reviewer for this valuable suggestion. We have revised our manuscript accordingly and made it more readable.

b. Discussion and conclusion section (line 306, page 14) “10 OTUs markers achieved ...” should be “10 OTU markers achieved ...”. Similar inconsistencies also appeared in Supplementary Table 4 and Table 5 (line 716 and 719, page 37).

Answer: We apologize for the spelling mistakes in the text. instead of OTUs, ASVs was selected in the revised version of our manuscript in order to better present our data. Thus, We've reanalyzed the data and avoided the same problem in the new results.

c. Statistical Analyses section (Line 379, page 17) mentioned “P or FDR values less than 0.05 were designated as significant differences (*, < 0.05, **, < 0.01, ***, < 0.001)”. The * that appears may be redundant.

Answer: We apologize for the mislabeling in the original manuscript. In the revised version of our manuscript, symbols (such as *) that represent statistical differences are used in **Figure 2A-B**. We have made corresponding corrections in figure legend (**Figure 2**).

d. Discussion and conclusion section (line 306, page 14) mentioned “which warrants

following mechanism investigations in vivo and vitro”. “in vivo and vitro” should be “in vivo and in vitro” and in italics.

Answer: We sincerely apologize for the spelling mistakes in the text. We have revised corresponding paragraph of our paper.

Reviewers' Comments:

Reviewer #2:

Remarks to the Author:

Thank you for addressing the feedback provided.

A few minor but important details to address before publication:

- Consider replacing the Chao1 diversity estimate with Observed count of ASVs as a measure of alpha diversity
- To avoid over-concluding, please remove 'diagnostic efficacy' or 'diagnostic potential'. These should be included in forward looking statements but not as part of the results. I suggest using 'to measure the strength of observed associations' or something similar. The use of 'biomarker' may be appropriate.
- Replace 'genus' with 'genera' when referring to more than one genus.
- For the results presented in Figure 3 and supplemental Figure 2, please include the differential features plots and make a comment about the prevalence of the bacteria identified as significant since LEfSe/LDA significance can be influenced by sparsity and also relative abundance.
- Figure 3 does not need the upstream taxonomic nomenclature provided to the left of the LEfSe plot. I suggest to adapt the LEfSe graph (remove vertical lines for clarity), and mark to the right of the plot with MaAsLin and metagenomic annotations.
- Similarly, only reporting genus-level or last identified taxon Figure 5B and Figure 6B would help with clarity.
- Consider running RF on the metabolic imputation/KEGG modules outputs to validate the strength of the taxonomic associations. I would suggest moving Figure 4 to supplemental or pick a handful of biologically relevant pathways to present.

Reviewer #3:

Remarks to the Author:

The paper is revised well. I'm satisfied.

Reviewer #4:

Remarks to the Author:

1- As suggested, the authors have provided additional information in order to compare microbiome signatures relative to tumor location (right vs left side) and stage (early vs stage IV) (previous comments #1 and 2).

2- However, the additional data provided on FIT in response to previous comment #3, which is the current screening test for colorectal cancer, in order to compare with the diagnostic value of their microbial markers, appear flawed. Indeed, FIT is now being used in most countries and is recognized to detect 75-80% of patients with a colorectal cancer with a specificity of 95% (see ref. 28) leading to an AUC of close to 0.90. With a rate of detection below 35% (supp Table1), the authors reach an AUC value varying between .50 and .59 which is considered to be well below acceptable values. The manufacturer identity of the kit used and cut-off values for Hg detection should be provided. Nevertheless, under these conditions, their claim that the microbial signatures are more accurate than FIT for colorectal cancer screening does not appear to be valid. I would suggest comparing with FIT data from the accepted literature.

3- In the context of establishing a diagnostic test, it would be pertinent to calculate the cut-off

values for the ROC curves to evaluate the sensitivities and specificities for the microbial markers in addition to the AUC data.

4- Another aspect pertains to the use of the expression "prospectively included" to describe the patient cases (line 122) while it appears that they were retrospectively classified as matched control and cancer following colonoscopy. The expression should therefore be omitted.

5- Finally, as mentioned, the detection of CRC in the population younger than 50 is a topic of great interest because of the growing incidence and aggressiveness, although there is no clear consensus yet on how to implement efficient and cost-controlled screening programs. In this context, it would be interesting for the authors to develop their conclusion about the potential of microbial markers for detecting CRC in the younger population.

Response Letter

REVIEWER COMMENTS

Reviewer #2 (Remarks to the Author):

Thank you for addressing the feedback provided.

A few minor but important details to address before publication:

1.- Consider replacing the Chao1 diversity estimate with Observed count of ASVs as a measure of alpha diversity

Answer: Thank you for your comment. We have made revisions (including the results and the figure) according to your suggestion (color coded, Figure 2A).

Figure 2

Figure 2. Bacterial diversity of the fecal microbiota associated with yCRC and oCRC

2.- To avoid over-concluding, please remove 'diagnostic efficacy' or 'diagnostic potential'. These should be included in forward looking statements but not as part of the results. I suggest using 'to measure the strength of observed associations' or something similar. The use of 'biomarker' may be appropriate.

Answer: Thank you very much for your suggestion. We have modified the description of diagnostic value in the methods and results (color coded) to avoid over-concluding according to your suggestion.

3.- Replace 'genus' with 'genera' when referring to more than one genus.

Answer: Thanks for the suggestion. We have made revisions throughout the manuscript according to your suggestion (color coded).

4.- For the results presented in Figure 3 and supplemental Figure 2, please include the differential features plots and make a comment about the prevalence of the bacteria identified as significant since LefSe/LDA significance can be influenced by sparsity and also relative abundance.

Answer: Thank you for your suggestion. We have supplemented the statistical distribution and relative abundance of these specific genus-level (16S rRNA sequencing) and species-level (metagenomics sequencing) biomarkers based on LefSe analysis, and the Max, Min, median, mean, SD, and SEM values supported the prevalence of the bacteria in most of the samples. We have added related descriptions in the result section (color coded) and Table S6- S9.

Table S6. The statistical distribution and relative abundance of the specific genus-level biomarkers for oCRC in Fudan cohort

Genus	oControl						oCRC					
	Max	Min	Median	Mean	SD	SEM	Max	Min	Median	Mean	SD	SEM
g_[Eubacterium]_coprostanoligenes_group	0.17499	0	0.002764	0.013364	0.028422	0.001995	0.091962	0	0.000397	0.009189	0.016546	0.001084
g_[Eubacterium]_eligans_group	0.10039	0	0.000364	0.004733	0.013165	0.000924	0.039212	0	0	0.001705	0.004621	0.000303
g_[Eubacterium]_ventriosus_group	0.035127	0	0.000313	0.001835	0.004493	0.000315	0.025373	0	0	0.000899	0.002741	0.00018
g_Agathobacter	0.409717	0	0.009215	0.030581	0.056628	0.003975	0.226527	0	0.000643	0.01443	0.033709	0.002208
g_Anaerostipes	0.154941	0	0.001386	0.007275	0.019419	0.001363	0.361837	0	0.000311	0.006133	0.02596	0.001701
g_Bifidobacteria	0.065115	0	0.000431	0.002608	0.006672	0.000468	0.062177	0	0.000148	0.00151	0.004736	0.00031
g_Blautia	0.168578	0	0.012229	0.020227	0.025407	0.001783	0.232009	0	0.009103	0.01718	0.027687	0.001814
g_Brevundimonas	0.063985	0	0	0.001055	0.005066	0.000356	0.1807	0	8.23E-05	0.001791	0.01213	0.000795
g_Butyricoccus	0.123327	0	0.001969	0.004596	0.012147	0.000853	0.025488	0	0.000521	0.002006	0.003593	0.000235
g_Christensenellaceae_R-7_group	0.189775	0	0.000209	0.008191	0.023323	0.001567	0.226188	0	0	0.007331	0.025905	0.001697
g_Clostridium_sensu_stricto_1	0.128872	0	0.000951	0.006508	0.01758	0.001234	0.220444	0	0.0002	0.005066	0.017557	0.00115
g_Collinsella	0.684017	0	0.001154	0.01237	0.054525	0.003827	0.231103	0	0	0.011254	0.029218	0.001914
g_Coprococcus	0.151496	0	0.002975	0.0081	0.016241	0.00114	0.113498	0	0.000569	0.005847	0.012733	0.000834
g_Eggerthella	0.21858	0	5.99E-05	0.001706	0.015388	0.00108	0.099529	0	0.000364	0.002327	0.00778	0.00051
g_Erysipelatoclostridium	0.503735	0	0	0.006358	0.03841	0.002696	0.44798	0	0.00023	0.0067	0.032629	0.002138
g_Erysipelotrichaceae_UCG-003	0.558735	0	0.002401	0.02627	0.077975	0.005473	0.402739	0	0.000475	0.019292	0.051775	0.003392
g_Escherichia-Shigella	0.851289	0	0.005836	0.057052	0.132201	0.009279	0.985596	0	0.01101	0.079291	0.15133	0.009914
g_Faecalibacterium	0.697016	0	0.062629	0.109628	0.118952	0.008349	0.632802	0	0.020687	0.061863	0.098059	0.006424
g_Fusicatenibacter	0.199136	0	0.002807	0.008634	0.01908	0.001339	0.068336	0	0.001031	0.005838	0.012019	0.000787
g_Fusobacterium	0.569904	0	0	0.008555	0.055485	0.003894	0.674828	0	0.000544	0.025113	0.073146	0.004792
g_Gemella	0.01021	0	0	0.000316	0.000976	6.85E-05	0.260458	0	0.000237	0.00618	0.025585	0.001676
g_Haemophilus	0.100783	0	0.000137	0.002383	0.00849	0.000596	0.330914	0	0	0.004475	0.025568	0.001675
g_Hungatella	0.223947	0	0	0.002754	0.021128	0.001483	0.187919	0	0.00028	0.005189	0.020193	0.001323
g_Incertae_Sedis	0.050495	0	0.000834	0.001945	0.004635	0.000325	0.106078	0	0.000421	0.00223	0.00852	0.000558
g_Lachnoclostridium	0.122833	0	0.009357	0.015664	0.020829	0.001462	0.176514	0	0.004849	0.01714	0.030943	0.002027
g_Lachnospira	0.126356	0	0.001346	0.01095	0.021116	0.001482	0.063473	0	0	0.00198	0.006604	0.000433
g_Lachnospiraceae_NK4A136_group	0.068464	0	0.001429	0.004522	0.00776	0.000545	0.067011	0	0	0.00383	0.008484	0.000556
g_Monoglobus	0.140034	0	0.00222	0.007699	0.017107	0.001201	0.096161	0	0.000275	0.003922	0.009701	0.000636
g_Parasutterella	0.222874	0	0.000137	0.004708	0.01803	0.001265	0.10962	0	0	0.002693	0.011169	0.000732
g_Parvimonas	0.004432	0	0	0.000119	0.000512	3.60E-05	0.233044	0	0.000288	0.008716	0.030082	0.001971
g_Peptostreptococcus	0.008747	0	0	0.000142	0.00087	6.11E-05	0.150852	0	0.000184	0.005462	0.018319	0.0012
g_Phascloartrobacterium	0.526131	0	0.00235	0.019114	0.054762	0.003844	0.134905	0	0.001115	0.01013	0.02107	0.00138
g_Romboutsia	0.379284	0	0.001678	0.009848	0.03379	0.002372	0.126022	0	0	0.003872	0.014781	0.000968
g_Roseburia	0.186	0	0.010301	0.020254	0.027096	0.001902	0.215107	0	0.001254	0.016292	0.031759	0.002081
g_Ruminococcus	0.413073	0	0.005339	0.020989	0.047882	0.003361	0.173679	0	0.000462	0.011426	0.027287	0.001788
g_Streptococcus	0.588478	0	0.003303	0.021163	0.058034	0.004073	0.648595	0	0.005241	0.041286	0.09877	0.006471
g_Subdoligranulum	0.283225	0	0.01492	0.034455	0.051638	0.003624	0.427594	0	0.003961	0.024358	0.049018	0.003211
g_UBA1819	0.021296	0	0.000338	0.000906	0.002054	0.000144	0.11483	0	0.000663	0.004484	0.013278	0.00087

Abbreviations: CRC, colorectal cancer; oControl, old-onset healthy control; oCRC, old-onset CRC; yCRC, young-onset CRC; yControl, young-onset healthy control; Max, maximum; Min, minimum; SD, standard deviation; SEM, standard error of mean

Table S9. The statistical distribution and relative abundance of the specific genus-level biomarkers (identified by metagenomic sequencing) for yCRC in Fudan cohort.

Genus	M_HY						M_Y					
	Max	Min	Median	Mean	SD	SEM	Max	Min	Median	Mean	SD	SEM
s__[Eubacterium]_rectale	0.122474	0.000116	0.012008	0.024527	0.030308	0.004286	0.086026	0.000129	0.004943	0.01442	0.021973	0.003108
s__Bacteroides_caecimuris	0.029473	8.11E-05	0.009487	0.011029	0.006714	0.00095	0.102775	0.004023	0.011032	0.016142	0.016254	0.002299
s__Bacteroides_ovatus	0.236875	0.000121	0.019712	0.035782	0.048692	0.006886	0.331672	0.006911	0.027974	0.052088	0.060141	0.008505
s__Clostridiales_bacterium_CCNA10	0.01559	0.000273	0.00341	0.004038	0.003374	0.000477	0.032944	0.000174	0.004883	0.006964	0.007422	0.00105
s__Desulfovibrio_piger	0.011613	5.64E-06	0.000128	0.000558	0.001744	0.000247	0.026575	1.59E-06	0.00017	0.001391	0.004085	0.000578
s__Eggerthella_lenta	0.00102	1.14E-05	0.000118	0.000169	0.00018	2.55E-05	0.003771	2.81E-05	0.00017	0.000379	0.000616	8.72E-05
s__Enterocloster_bolteae	0.049358	0.000491	0.002215	0.00467	0.007588	0.001073	0.201483	0.000169	0.003584	0.01044	0.028688	0.004057
s__Flavonifractor_plautii	0.020455	0.000204	0.00252	0.004104	0.004438	0.000628	0.033692	0.000162	0.004383	0.007083	0.007307	0.001033
s__Lachnospira_eligens	0.076135	5.31E-05	0.006845	0.013953	0.016734	0.002367	0.081137	5.49E-05	0.003175	0.01125	0.01779	0.002516
s__Oscillibacter_sp_PEA192	0.012748	4.28E-05	0.002084	0.002737	0.002481	0.000351	0.026806	2.79E-05	0.00358	0.004947	0.005305	0.00075
s__Roseburia_intestinalis	0.096304	0.000225	0.006062	0.011681	0.015685	0.002218	0.027164	0.00022	0.004283	0.005034	0.004614	0.000653
s__Ruthenibacterium_lactatiformans	0.017055	9.50E-05	0.002006	0.002609	0.002632	0.000372	0.06659	8.47E-05	0.002668	0.00496	0.00949	0.001342

Abbreviations: M_HY, young-onset healthy control; M_Y, young-onset colorectal cancer; Max, maximum; Min, minimum; SD, standard deviation; SEM, standard error of mean

5.- Figure 3 does not need the upstream taxonomic nomenclature provided to the left of the LefSe plot. I suggest to adapt the LefSe graph (remove vertical lines for clarity), and mark to the right of the plot with MaAsLin and metagenomic annotations.

Answer: Thank you for the valuable advice. We have removed the redundant taxonomic nomenclature and vertical lines, and added a column of species taxonomy based on metagenomic sequencing data on the right-side, which will more intuitively reflect the correspondence between 16s rRNA annotations and metagenomic annotations (Figure 3).

Figure 3. Phylogenetic profiles of fecal microbial communities in yCRC and oCRC.

6.- Similarly, only reporting genus-level or last identified taxon Figure 5B and Figure 6B would help with clarity.

Answer: Thank you for your comment. We have removed the redundant taxonomic name according to your suggestion (Figure 4B, Figure 5B). The Figure 4 has been moved to supplemental, thus, the serial numbers of the Figures and Tables in the text have been adjusted accordingly.

Figure 4

Figure 4. Identification of microbial markers of oCRC by random forest models.

Figure 5

Figure 5. Identification of microbial markers of yCRC using random forest models.

7.- Consider running RF on the metabolic imputation/KEGG modules outputs to validate the strength of the taxonomic associations. I would suggest moving Figure 4 to supplemental or pick a handful of biologically relevant pathways to present.

Answer: Thank you for your valuable advice. Firstly, we regard KEGG based metabolic imputation as a prediction, which has potential reference value. However, the metabolic process of tumor and microbe interaction still needs to be further validated by metabolomics and tumor biology experiments. In addition, the modules annotated by KEGG are still too rough. We expected to get more accurate annotations with the support of comprehensive database, more advanced sequencing technologies, and multi-omics analysis in the future. Therefore, we consider metabolic imputation as a supplement to the bacteria spectrum of young-onset and old-onset CRC.

Accordingly, we have moved the Figure 4 to the supplementary materials (Figure S3) according to your suggestion.

Reviewer #3 (Remarks to the Author):

The paper is revised well. I'm satisfied.

Answer: Thank you for your encouraging comments.

Reviewer #4 (Remarks to the Author):

1- As suggested, the authors have provided additional information in order to compare microbiome signatures relative to tumor location (right vs left side) and stage (early vs stage IV) (previous comments #1 and 2).

Answer: Thank you for your encouraging comments.

2- However, the additional data provided on FIT in response to previous comment #3, which is the current screening test for colorectal cancer, in order to compare with the diagnostic value of their microbial markers, appear flawed. Indeed, FIT is now being used in most countries and is recognized to detect 75-80% of patients with a colorectal cancer with a specificity of 95% (see ref. 28) leading to an AUC of close to 0.90. With a rate of detection below 35% (supp Table1), the authors reach an AUC value varying between .50 and .59 which is considered to be well below acceptable values. The manufacturer identity of the kit used and cut-off values for Hg detection should be provided. Nevertheless, under these conditions, their claim that the microbial signatures are more accurate than FIT for colorectal cancer screening does not appear to be valid. I would suggest comparing with FIT data from the accepted literature.

Answer: We very much appreciate this comment. We apologize for this incorrect description of the fecal occult blood test (FOBT) requested (previous comments #1) in the revised manuscript. The detection method of fecal occult blood used in our study is qualitative FOBT rather than quantitative FIT. In this study, the data of subjects' fecal occult blood test was collected through clinical data. Therefore, we investigated the fecal occult blood test method in the hospital's laboratory. As shown in the Response Letter Figure 1, Figure 2 and Figure 3. In the clinical laboratory of our hospital, whether it is the colloidal gold method to detect fecal hemoglobin or the immune-chromatographic method to detect fecal transferrin, both the results of the test are qualitative rather than quantitative. We again apologize for the error description in the previous text. Our research is mainly to explore the recognition of the random forest model for CRC in young people and CRC in the elderly. As the reviewers mentioned in the first revision: "It would be useful if the diagnostic accuracy of the microbial signature can be compared to the faecal occult blood test which is the current screening test." we compared the diagnostic efficacy of microbial markers with the conventional non-invasive and invasive clinical diagnostic methods fecal FOBT and serum CEA and CA19-9. After consulting the published literature¹⁻⁸ (The references are shown below), we found that the diagnostic accuracy of FOBT in the published literature ranged

from 25.5% to 86.3% with sensitivity and specificity ranged from 7.4%–75.0% and 21.6%–98.6%, respectively. Our FOBT results reached an AUC value varying between 0.50 and 0.62, which was comparable to the published literature. Moreover, our results suggest the potential of using faecal microbiome to develop a specific diagnostic test for yCRC or oCRC. yCRC and oCRC groups have their unique fecal microbial markers for better diagnostic accuracy. We have modified the results and Figure accordingly.

Reference for Response Letter:

1. Ladabaum U, Dominitz JA, Kahi C, Schoen RE. Strategies for Colorectal Cancer Screening. *Gastroenterology*. 2020;158(2):418-432. doi: 10.1053/j.gastro.2019.06.043.
2. Shapiro JA, Bobo JK, Church TR, et al. A Comparison of Fecal Immunochemical and High-Sensitivity Guaiac Tests for Colorectal Cancer Screening. *The American journal of gastroenterology*. 2017; 112(11): 1728–35. DOI: 10.1038/ajg.2017.285.
3. Jucá MJ, Assunção PRT and Hasten-Reiter Júnior HN. Fecal occult blood test and flexible rectosigmoidoscopy: Tools for the screening of colorectal neoplasms in asymptomatic patients. *Journal of Coloproctology*. 2015; 35(1): 35–41. DOI: 10.1016/j.jcol.2015.01.002.
4. Elsafi SH, Alqahtani NI, Zakary NY and Al Zahrani EM. The sensitivity, specificity, predictive values, and likelihood ratios of fecal occult blood test for the detection of colorectal cancer in hospital settings. *Clinical and Experimental Gastroenterology*. 2015; 8: 279–84. DOI: 10.2147/CEG.S86419.
5. Lohsiriwat V. Accuracy of self-checked fecal occult blood testing for colorectal cancer in Thai patients. *Asian Pacific Journal of Cancer Prevention*. 2014; 15(18): 7981–4. DOI: 10.7314/APJCP.2014.15.18.7981.
6. Redwood D, Provost E, Asay E, et al. Comparison of fecal occult blood tests for colorectal cancer screening in an Alaska native population with high prevalence of *Helicobacter pylori* infection, 2008–2012. *Preventing Chronic Disease*. 2014; 11(4). DOI: 10.5888/pcd11.130281.
7. Yeasmin F, Ali M, Rahman M, Sultana T, Rahman MQ and Ahmed A. A comparative study of chemical and immunological method of fecal occult blood test in the diagnosis of occult lower gastrointestinal bleeding. *Bangladesh Medical Research Council Bulletin*. 2014; 39(2): 52–6. DOI: 10.3329/bmrcb.v39i2.19641.
8. Ramdzan AR, Abd Rahim MA, Mohamad Zaki A, Zaidun Z, Mohammed Nawi A. Diagnostic Accuracy of FOBT and Colorectal Cancer Genetic Testing: A Systematic Review & Meta-Analysis. *Ann Glob Health*. 2019;85(1):70. Published 2019 May 15. doi:10.5334/aogh.2466

[Redacted]

Response Figure 1. Fecal occult blood (FOB) diagnostic kit (Colloidal Gold) to detect fecal hemoglobin. Figure 1A and 1B, The first page of the package cover and the second page of the package cover of the fecal occult blood test kit. Figure 1C and 1D, Product Manual.

* Product name: Fecal Occult Blood (FOB) Diagnostic Kit (Colloidal Gold) .

** Purpose: This detection reagent is used to qualitatively detect hemoglobin in human feces in vitro.

*** Detection principle: This detection kit uses immunogold technology and double-antibody sandwich method to detect hemoglobin in human feces. When the sample contains hemoglobin, it will react with the colloidal gold-labeled HB2 monoclonal antibody to form a complex. The reaction complex moves forward along the nitrocellulose membrane to the detection area under the action of chromatography. HB1 monoclonal antibody capture, a red reaction line appears in the detection area, and the result is positive at this time. When there is no human hemoglobin in the sample, there is no red reaction line in the detection area, and the result is negative. Regardless of whether the sample contains hemoglobin, the quality control line will show color.

**** Test results and interpretation: Positive (A red line appears at the position of the test line and the quality control line of the test reagent card, indicating that there is occult blood in the sample); Negative (Only a red line appears at the position of the quality control line, indicating that there is no occult blood in the sample); Invalid (No red line appears on the position of the quality control line, indicating that the result is invalid).

***** Limitations of detection methods: This detection reagent is used to qualitatively detect hemoglobin in human feces in vitro, but it cannot be quantified. A positive result only indicates the presence of human hemoglobin in the stool, but it cannot distinguish the cause. The results of this test kit should be judged by the doctor in conjunction with clinical and other examinations and clinical symptoms. A negative result does not rule out bleeding. When the sample concentration is higher than 2mg/ml, a hook effect may occur.

***** Basic Information: Manufacturer: Zhuhai Keyu Biological Engineering Co.,Ltd. Address:

2nd Floor, Building D, Kindly International Medical Industrial Park, No. 288, Airport Road East, Jinwan District, Zhuhai City, GuangDong Province. Email: export@keyubio.com. Website: <http://www.en.keyubio.com>.

[Redacted]

Response Figure 2. Transferrin (Tf) test kit (Immunochromatography) to detect fecal transferrin. Figure 2A and 2B, The first page of the package cover and the second page of the package cover of the transferrin (Tf) test kit. Figure 2C and 2D, Product Manual.

Product name: Transferrin (Tf) Test Kit (Immunochromatography).

Purpose: This reagent is suitable for the qualitative detection of transferrin in human fecal samples. It is mainly used for the clinical evaluation of gastrointestinal bleeding and is only used for in vitro diagnosis.

Detection principle: The transferrin detection kit uses colloidal gold immunochromatography to qualitatively detect transferrin in human feces. The sample containing human transferrin is combined with monoclonal antibody, and a red band appears in the detection area (T), and the result is positive; if there is no human transferrin in the sample, there is no red band in the detection area (T). The result was negative. The remaining colloidal gold antibody-antigen complex is chromatographed to the quality control area (C), and a red band will appear in the quality control area (C).

Test results and interpretation: When a red band appears at both the detection area (C) and the quality control area (T), the result is positive; when only a red band appears at the quality control area (C), the result is negative; when there is no red band at the quality control area (C), it means that the test card is invalid.

Limitations of detection methods: This reagent can only be used for qualitative detection but not quantitative detection. This reagent cannot make a conclusive diagnosis of gastrointestinal bleeding lesions. Limited by the detection range, negative results may be caused by too low or too high human transferrin concentration

Basic Information: Zhuhai Encode Medical Engineering Co., Ltd

Add:NO.020,Honghui 2nd RD Hongqi Industrial Zone,Jiwan District,Zhuhai,P.R China(519090).

Email: encode01@hotmail.com / itd@encode.com.cn. Tel: +86-756-3981528

Fax:+86-756-3983809. Tech. Support:+86-756-3983818. Website: <http://www.encode.com.cn/en>

[Redacted]

Response Figure 3. Automatic stool analyzer detection system. Figure 3A and 3B, Automatic stool analyzer detection system used in the hospital's laboratory in our study (KU-F10, Fully automatic feces analyzer. http://en.keyubio.com/portal/article/index/cid/11/id/67/is_product/1.html). Figure 3C, Stool container. Figure 3D, Card type of stool occult blood detection kit. Figure 3E, information system for fully automatic stool analyzer KU-F10. Figure 3F, a representative negative sample: The test results of colloidal gold method (FOB) and immunochromatographic method (TF) are both negative. Figure 3G, a representative positive sample: The test results of colloidal gold method (FOB) and immunochromatographic method (TF) are both positive.

3- In the context of establishing a diagnostic test, it would be pertinent to calculate the cut-off values for the ROC curves to evaluate the sensitivities and specificities for the microbial markers in addition to the AUC data.

Answer: Thank you for this useful suggestion. We have supplemented the cut-off values for the ROC curves in Table S12 and Table S21.

Table S12. The best cut-off values of the ROC curves for fecal microbial identified markers by random forest models.

Cohort	Group	Best cut-off value	Specificity	Sensitivity	AUC
Training set from Fudan cohort	-	-	-	-	-
	oCRC vs. oControl	0.535	0.83	0.824	0.8928
	yCRC vs. yControl	0.583	0.895	0.685	0.8657
Testing set from Fudan cohort	-	-	-	-	-
	oCRC vs. oControl	0.558	0.857	0.765	0.8667
	yCRC vs. yControl	0.471	0.674	0.778	0.7952
Validation set from Huadong cohort	-	-	-	-	-
	oCRC vs. oControl	0.47	0.959	0.722	0.8667
	yCRC vs. yControl	0.285	0.849	0.852	0.878

Abbreviations: CRC, colorectal cancer; oControl, old-onset healthy control; oCRC, old-onset CRC; yCRC, young-onset CRC; yControl, young-onset healthy; AUC, area under receiving operating characteristics curve; ROC, receiver operating curve

Table S21. The best cut-off values for the ROC curves using different markers.

Cohort	Group	Best cut-off value	Specificity	Sensitivity	AUC
Fudan cohort	-	-	-	-	-
oCRC vs. oControl	-	-	-	-	-
	Microbial markers	0.606	0.892	0.773	0.8961
	FOBT	0.032	0.985	0.33	0.5833
	CEA	0.995	0.995	0.468	0.7439
	CA199	0.86	0.833	0.468	0.6671
	CEA + CA199	0.643	0.877	0.648	0.8199
yCRC vs. yControl	-	-	-	-	-
	Microbial markers	0.465	0.682	0.854	0.8511
	FOBT	0.011	0.973	0.333	0.5566
	CEA	0.709	0.851	0.576	0.7711
	CA199	0.885	0.865	0.438	0.6984
	CEA + CA199	0.445	0.77	0.778	0.8254
Huadong cohort	-	-	-	-	-
oCRC vs. oControl	-	-	-	-	-
	Microbial markers	0.361	0.89	0.778	0.8656
	FOBT	0.04	0.365	0.944	0.5076
	CEA	0.068	0.603	0.889	0.7437
	CA199	0.062	0.563	0.852	0.6721
	CEA + CA199	0.134	0.548	0.926	0.7851
yCRC vs. yControl	-	-	-	-	-
	Microbial markers	0.727	0.902	0.681	0.8561
	FOBT	0.502	0.305	1	0.6203
	CEA	0.326	0.561	0.855	0.7296
	CA199	0.968	0.78	0.377	0.5832
	CEA + CA199	0.41	0.512	0.899	0.7252

Abbreviations: CRC, colorectal cancer; oControl, old-onset healthy control; oCRC, old-onset CRC; yCRC, young-onset CRC; yControl, young-onset healthy; AUC, area under receiving operating characteristics curve; FOBT, fecal occult blood test; ROC, receiver operating curve; Microbial markers were identified by random forest models.

4- Another aspect pertains to the use of the expression "prospectively included" to describe the patient cases (line 122) while it appears that they were retrospectively classified as matched control and cancer following colonoscopy. The expression should therefore be omitted.

Answer: These modifications have been made to the revised manuscript.

5- Finally, as mentioned, the detection of CRC in the population younger than 50 is a topic of great interest because of the growing incidence and aggressiveness, although there is no clear consensus yet on how to implement efficient and cost-controlled screening programs. In this context, it would be interesting for the authors to develop their conclusion about the potential of microbial markers for detecting CRC in the younger population.

Answer: Thank you for your encouraging comments.

Reviewers' Comments:

Reviewer #4:

Remarks to the Author:

1- The additional information provided about the hemoglobin detection method used allows a better understanding of the poor rate of CRC detection reported in the work with this non-quantitative method. This method is clearly not an international standard test for CRC screening and while proposing a new diagnosis test for CRC, the claim that "this signature showed better predicting performance than fecal occult blood test and traditional serum tumor markers " (lines 77-78) is biased. Comparison with the international standards is mandatory and it is the FIT (or iFOBT) that is in use in most countries and that should be considered for this purpose.

1A- While I understand that the patients in this study have not been tested for FIT, data are available from the literature (as mentioned in my last comments) and should be used as a reference to discuss and compare the potential of the gut microbiota biomarker-based test presented.

1B-Furthermore, to avoid any confusion, the fecal occult blood test used in this work should be clearly identified as "non-quantitative FOBT" everywhere.

2- As mentioned in the last remark of my previous evaluation, the detection of CRC in the population younger than 50 is a topic of great interest due to growing incidence and aggressiveness, although there is no clear consensus yet on how to implement efficient and cost-controlled screening programs. In this context, the authors should develop their conclusion about the potential of microbial markers for detecting CRC in the younger population.

Reviewer #5:

Remarks to the Author:

This is an interesting study with a well-thought design and a large enough sample size. Only drawback is the use of 16S rRNA gene sequencing instead of shotgun metagenomics.

Overall I think the authors replied appropriately to the issues that were raised by the reviewers and the paper looks solid. There are some parts in which the language is not very smooth or a bit imprecise, but it can probably be improved at proofreading.

One aspect that is not acceptable at all is the fact that the data does not seem to be available and the statement "All the data related to this paper may be requested from the authors" is very concerning. All the sequencing data (16S and metagenomics) should be deposited in a public repository with appropriate metadata both in the repository and as supplementary material of the paper. Not providing the data publicly is against the policies of Nature journals and goes against the authors themselves as the results are not replicable and cannot be reused for additional (meta)analyses.

Some minor remarks:

- The authors should not use the term "flora" to refer to the microbiota or microbiome
- when defining yCRC oCRC etc in the Results, the authors should report the details about the age ranges etc
- Line 157 "significantly separated distribution". The distributions are overlapping, not separated. In case is a "significantly different distribution"
- Typos:
 - Line 232 "To evaluated the classification power"
 - Line 268 "In clinical, FOBT"

REVIEWER COMMENTS

Reviewer #4 (Remarks to the Author):

1- The additional information provided about the hemoglobin detection method used allows a better understanding of the poor rate of CRC detection reported in the work with this non-quantitative method. This method is clearly not an international standard test for CRC screening and while proposing a new diagnosis test for CRC, the claim that "this signature showed better predicting performance than fecal occult blood test and traditional serum tumor markers " (lines 77-78) is biased. Comparison with the international standards is mandatory and it is the FIT (or iFOBT) that is in use in most countries and that should be considered for this purpose.

Answer: Thank you very much for pointing it out and we are aware of the limitation. As recommended, We have removed the controversial statements in the "Abstract/Results/Discussion" sections and we compared the FIT and non-quantitative FOBT data published in the literature with our fecal microbial data, and further discussed and cited them in the "Discussion" section. We have revised our conclusions in the "Abstract/Results/Discussion" sections to make the conclusions more rigorous and understandable according to your valuable suggestion.

1A- While I understand that the patients in this study have not been tested for FIT, data are available from the literature (as mentioned in my last comments) and should be used as a reference to discuss and compare the potential of the gut microbiota biomarker-based test presented.

Answer: Thank you for your valuable advice. We agree that this is truly an important point. According to your suggestion, we discussed the use of published FIT and non-quantitative FOBT data in the diagnosis of CRC in the "Discussion" section. We also evaluated the fecal microbiota results in our study with the published FIT data in the literature. Please see the following in our revised manuscript:

In addition to the fecal microbiota that has received increasing attention, FOBT, including either guaiac-based (gFOBT, non-quantitative FOBT) or immunological-based (iFOBT, FIT), is also a non-invasive method and is currently implemented for CRC screening in average-risk population^{48, 49, 50}. Specifically, non-quantitative FOBT detects hemoglobin, while quantitative FIT detects globin and is not affected by diet⁵¹. The diagnostic accuracy of non-quantitative FOBT in the published literature ranged from 25.5% to 86.0% with sensitivity and specificity values ranged from 7.4%–75.0% and 21.6%–98.6%, respectively^{52, 53, 54, 55, 56}. FIT for hemoglobin shows better CRC diagnostic performance than non-quantitative FOBT⁵⁷. However, its high diagnostic sensitivity comes at the cost of a high false positive rate^{58, 59}. Furthermore, the optimal threshold of FIT for CRC screening is still unknown because the fecal hemoglobin concentration varies with gender and age⁶⁰. A systematic review with meta-analysis reported that FIT had a pooled sensitivity of 91% and a pooled specificity of 90% in detecting CRC using a positivity threshold of 10 µg/g, whereas a threshold ≥ 20 µg/g resulted in a pooled sensitivity of 75% and a pooled specificity of 0.95⁶¹. This review is highly heterogeneous by population setting, age range, FIT brand and FIT threshold used. In a primary care population with low-risk symptoms of CRC, the AUC for the FIT was 0.92⁶². Another review further quantified the performance characteristics of FIT for CRC stratified by age, with a pooled sensitivity of 85% for ages 50 to 59 and a sensitivity of 73% for ages 60 to 69, but did not compare the diagnostic efficacy of FIT in

young people under 50⁶³ In our study, the sensitivity and specificity for the detection of CRC with the gut microbiota biomarkers were comparable with non-quantitative FOBT but inferior to FIT data published in the above-mentioned literature. However, our study not only emphasizes the use of gut microbiota biomarkers as a promising non-invasive tool to detect CRC, but we also highlights its use to distinguish yCRC and oCRC. The potential precision screening effect of fecal microbiota on yCRC provide a unique opportunity for clinical practice, which may help early identification of more individuals with yCRC.

References:

- 1) Hewitson P, Glasziou P, Watson E, Towler B, Irwig L. Cochrane systematic review of colorectal cancer screening using the fecal occult blood test (hemoccult): an update. *The American journal of gastroenterology* 103, 1541-1549 (2008).
- 2) Timmouth J, Lansdorp-Vogelaar I, Allison JE. Faecal immunochemical tests versus guaiac faecal occult blood tests: what clinicians and colorectal cancer screening programme organisers need to know. *Gut* 64, 1327-1337 (2015).
- 3) Robertson DJ, et al. Recommendations on Fecal Immunochemical Testing to Screen for Colorectal Neoplasia: A Consensus Statement by the US Multi-Society Task Force on Colorectal Cancer. *Gastroenterology* 152, 1217-1237 e1213 (2017).
- 4) Ramdzan AR, Abd Rahim MA, Mohamad Zaki A, Zaidun Z, Mohammed Nawi A. Diagnostic Accuracy of FOBT and Colorectal Cancer Genetic Testing: A Systematic Review & Meta-Analysis. *Ann Glob Health* 85, (2019).
- 5) Shapiro JA, et al. A Comparison of Fecal Immunochemical and High-Sensitivity Guaiac Tests for Colorectal Cancer Screening. *The American journal of gastroenterology* 112, 1728-1735 (2017).
- 6) Elsafi SH, Alqahtani NI, Zakary NY, Al Zahrani EM. The sensitivity, specificity, predictive values, and likelihood ratios of fecal occult blood test for the detection of colorectal cancer in hospital settings. *Clin Exp Gastroenterol* 8, 279-284 (2015).
- 7) Lohsiriwat V. Accuracy of self-checked fecal occult blood testing for colorectal cancer in Thai patients. *Asian Pacific journal of cancer prevention : APJCP* 15, 7981-7984 (2014).
- 8) Redwood D, et al. Comparison of fecal occult blood tests for colorectal cancer screening in an Alaska Native population with high prevalence of *Helicobacter pylori* infection, 2008-2012. *Prev Chronic Dis* 11, E56 (2014).
- 9) Yeasmin F, Ali MA, Rahman MA, Sultana T, Rahman MQ, Ahmed AN. A comparative study of chemical and immunological method of fecal occult blood test in the diagnosis of occult lower gastrointestinal bleeding. *Bangladesh Med Res Counc Bull* 39, 52-56 (2013).
- 10) Force USPST, et al. Screening for Colorectal Cancer: US Preventive Services Task Force Recommendation Statement. *JAMA* 325, 1965-1977 (2021).
- 11) Toes-Zoutendijk E, et al. Real-Time Monitoring of Results During First Year of Dutch Colorectal Cancer Screening Program and Optimization by Altering Fecal Immunochemical Test Cut-Off Levels. *Gastroenterology* 152, 767-775 e762 (2017).
- 12) Force USPST, et al. Screening for Colorectal Cancer: US Preventive Services Task Force Recommendation Statement. *JAMA* 315, 2564-2575 (2016).

- 13) Imperiale TF, Gruber RN, Stump TE, Emmett TW, Monahan PO. Performance Characteristics of Fecal Immunochemical Tests for Colorectal Cancer and Advanced Adenomatous Polyps: A Systematic Review and Meta-analysis. *Ann Intern Med* 170, 319-329 (2019).
- 14) Duran-Sanchon S, et al. Identification and Validation of MicroRNA Profiles in Fecal Samples for Detection of Colorectal Cancer. *Gastroenterology* 158, 947-957 e944 (2020).
- 15) Bailey SER, et al. Diagnostic performance of a faecal immunochemical test for patients with low-risk symptoms of colorectal cancer in primary care: an evaluation in the South West of England. *British journal of cancer* 124, 1231-1236 (2021).
- 16) Selby K, et al. Effect of Sex, Age, and Positivity Threshold on Fecal Immunochemical Test Accuracy: A Systematic Review and Meta-analysis. *Gastroenterology* 157, 1494-1505 (2019).

1B-Furthermore, to avoid any confusion, the fecal occult blood test used in this work should be clearly identified as "non-quantitative FOBT" everywhere.

Answer: Thanks for your suggestions and corrections. We have modified the results, tables and Figures in the revised version of our manuscript accordingly.

2- As mentioned in the last remark of my previous evaluation, the detection of CRC in the population younger than 50 is a topic of great interest due to growing incidence and aggressiveness, although there is no clear consensus yet on how to implement efficient and cost-controlled screening programs. In this context, the authors should develop their conclusion about the potential of microbial markers for detecting CRC in the younger population.

Answer: Thank you for your comments. Regarding the value of using gut microbiota to detect colorectal cancer in young people, we have revised our conclusions in the "Abstract/Discussion" to make the conclusions more rigorous according to your suggestion.

Please see revised conclusions in the section of "Abstract": This study suggests the association between gut microbiome dysbiosis and CRC risks in young groups, and highlights the potential of the gut microbiota biomarkers as a promising non-invasive tool for the accurate detection and distinction of individuals with yCRC.

Please see revised conclusions in the section of "Discussion": In conclusions, our current study revealed the common state of fecal microbial dysbiosis in yCRC and oCRC populations. Fecal microbiota-based biomarkers had a robust strength in distinguishing yCRC from aged-matched control. Although more clinical validations and mechanism investigations are needed, our study emphasizes the need to further study the potential association between the gut microbiota and the risk of CRC in young people, which may drive the clinical transformation of microbiota-based strategies into precision screening and diagnosis.

Reviewer #5 (Remarks to the Author):

This is an interesting study with a well-thought design and a large enough sample size. Only drawback is the use of 16S rRNA gene sequencing instead of shotgun metagenomics. Overall I think the authors replied appropriately to the issues that were raised by the reviewers and the paper looks solid. There are some parts in which the language is not very smooth or a bit imprecise, but it can probably be improved at proofreading.

Answer: Thank you for your encouraging comments and kind advice. Our paper have been edited by a professional editing company to improve the writing in our revised manuscript.

One aspect that is not acceptable at all is the fact that the data does not seem to be available and the statement "All the data related to this paper may be requested from the authors" is very concerning. All the sequencing data (16S and metagenomics) should be deposited in a public repository with appropriate metadata both in the repository and as supplementary material of the paper. Not providing the data publicly is against the policies of Nature journals and goes against the authors themselves as the results are not replicable and cannot be reused for additional (meta)analyses.

Answer: Thank you for your valuable suggestion. All the 16S rRNA gene sequences and metagenomic sequences data are available in the Sequence Read Archive repository. All data that support the findings of this study are available within the paper and the supplementary information files.

We have added the "Data Availability Statement" in the "Materials and Methods" accordingly: The 16S rRNA gene sequences data and the metagenomic sequences data that support the findings of this study were provided and publicly available at the NIH National Center for Biotechnology Information Sequence Read Archive (SRA) with BioProject ID PRJNA763023. Source data (Table S1 - S21) and the accession codes (Table S22 - S23) are provided in the supplementary material.

Some minor remarks:

- The authors should not use the term "flora" to refer to the microbiota or microbiome

Answer: Thank you for your advice. These modifications have been made to the revised manuscript.

- when defining yCRC oCRC etc in the Results, the authors should report the details about the age ranges etc

Answer: Thanks for the suggestion. We have supplemented the age ranges in the Results/Table S1/ Table S2/Table S3 accordingly.

Please see revised data in the "Results": Specifically, the Fudan cohort included 728 patients, with mean age of 63.23 ± 8.56 (25% - 75% percentile, 55 - 69) years in oControl, 64.26 ± 8.68 (25% - 75% percentile, 57 - 70) years in oCRC, 39.76 ± 6.11 (25% - 75% percentile, 35 - 45) years in yControl, and 40.45 ± 7.02 (25% - 75% percentile, 36 - 46) years in yCRC, respectively. The

Huadong cohort included 310 patients, with mean age of 60.46 ± 6.94 (25% - 75% percentile, 54 - 65) years in oControl, 62.42 ± 7.67 (25% - 75% percentile, 55 - 68) years in oCRC, 37.74 ± 6.19 (25% - 75% percentile, 33 - 42) years in yControl, and 39.68 ± 7.11 (25% - 75% percentile, 33 - 45) years in yCRC, respectively.

- Line 157 "significantly separated distribution". The distributions are overlapping, not separated. In case is a "significantly different distribution"

- Typos:

-- Line 232 "To evaluated the classification power"

-- Line 268 "In clinical, FOBT"

Answer: Thanks for pointing out the mistakes. The modifications have been made in the revised version of our manuscript.

Reviewers' Comments:

Reviewer #4:

Remarks to the Author:

The authors have adequately addressed my remarks in this revised version of their manuscript.